# Synchronous Home-Based Telerehabilitation of the Upper Extremity Following Stroke—A Pyramid Review

**DOI:** 10.3390/healthcare13010090

**Published:** 2025-01-06

**Authors:** Kirsten Stangenberg-Gliss, Christian Kopkow, Bernhard Borgetto

**Affiliations:** 1Faculty of Social Work and Health, University of Applied Sciences and Arts Hildesheim/Holzminden/Göttingen, 31134 Hildesheim, Germany; bernhard.borgetto@hawk.de; 2BG Hospital Berlin, 12683 Berlin, Germany; 3Department of Therapy Science I, Brandenburg University of Technology Cottbus-Senftenberg, 01968 Senftenberg, Germany; christian.kopkow@b-tu.de

**Keywords:** stroke, telerehabilitation, home based, Pyramid Review, evidence-based practice

## Abstract

Background: Stroke is a leading cause of long-term disability, often resulting in upper extremity impairment. Telerehabilitation offers a promising approach to deliver therapy in home settings. This review aimed to evaluate the effects of home-based telerehabilitation interventions delivered to address upper extremity function in stroke patients. Methods: A systematic review was conducted following the Pyramid Review methodology. Quantitative and qualitative studies examining home-based telerehabilitation for upper extremity function in stroke patients were included. Data were synthesized using meta-analysis where possible and narrative synthesis. Results: Thirty studies (24 quantitative, four qualitative, and two mixed-methods studies) were included. Telerehabilitation interventions demonstrated improvements in upper extremity function for both subacute and chronic stroke patients, with varying effect sizes across intervention types. Constraint-induced movement therapy adaptations and some sensor-based approaches showed consistently positive results. Qualitative findings revealed generally positive effects, with convenience, gamification, and social support as key motivators for adherence. Conclusions: Home-based telerehabilitation shows promise for improving upper extremity function in stroke patients. However, heterogeneity in intervention designs and outcomes limits definitive conclusions. Future research should focus on larger trials, observational studies, standardized outcome measures, and long-term follow-up as well as qualitative studies with focus on perceived effectiveness to optimize telerehabilitation approaches for stroke recovery.

## 1. Introduction

Stroke is one of the leading causes of long-term disability worldwide with significant impact on the quality of life of patients and their family members or caregivers [1]. Telerehabilitation interventions have emerged as crucial strategies to facilitate recovery and improve function after a stroke [2]. Nikolaev et al. defined telerehabilitation (TR) as ”the provision of rehabilitation services via telemedicine, i.e., using information and communication technologies, including video/teleconferencing, remote data collection equipment, telemonitoring, computers, mobile phones, robotics devices, exergames, and virtual reality (VR) tailored at individuals with disabilities, their families, clinicians, supervisors, and the community” [3]. Telerehabilitation can be provided both asynchronously and synchronously. Asynchronous telerehabilitation allows the patient to perform independent therapy and later review with the therapist. This is also known as “store and forward”. Synchronous telerehabilitation allows patients to engage in therapy sessions with their healthcare providers in real time, offering flexibility and accessibility that traditional rehabilitation may lack. Research indicates that synchronous telerehabilitation can lead to comparable outcomes to in-person sessions, promoting motor recovery and enhancing overall upper extremity function [4]. Moreover, this modality may reduce travel-related stress and ensure adherence to rehabilitation programs, further supporting long-term recovery [5]. As such, an evaluation of its effectiveness and efficacy is essential to understand its potential benefits fully. Therefore, we performed a Pyramid Review [6] to assess the current literature on synchronous home-based telerehabilitation for upper extremity functional impairments post-stroke. For the collection and synthesis of research findings in order to answer clinical questions about evidence-based decision-making in daily health care, a Pyramid Review offers a review approach that values and integrates external evidence from multiple research approaches.

Evidence-based medicine is coming under increasing criticism, not only, but also, because therapeutic decisions often cannot be guided primarily by randomised controlled trials (RCTs) [7,8,9]. Systematic reviews are often based on traditional evidence hierarchies and usually summarize RCTs only or rank them at the highest evidence level. Of course, there is a reason why RCTs are regarded as a gold standard: this type of study maximizes internal validity. However, the external validity of RCTs is limited due to the standardization and idealization of the intervention and the study conditions. To achieve both internally and externally valid study results, it is necessary to include observational studies as well as experimental studies in reviews. Conditions from experimental studies cannot simply be transferred to the everyday reality of care in the field. If the causes for the lack of transferability are not known (or cannot be adapted), the interventions studied experimentally cannot be recommended without any further ado for the provision of care. The more complex the interventions investigated, the greater this problem becomes.

To avoid possible misunderstandings: This conceptualization of experimental refers primarily to the interventions and their contexts (degree of innovation and manipulation), but not to the logic of experimental study designs in the sense of studies with an intervention group (experimental group) and a control group, nor does it refer to experimental preclinical animal and laboratory research.

Furthermore, quantitative studies reduce complex interactions in therapeutic practice to individual, decontextualized mechanisms of action. Re-contextualization in complex practice is usually left to the health professionals, often without further support from adequate evidence. However, members of the Health Services and Public Health Research Board of the UK Medical Research Council (MRC) consider qualitative studies to be appropriate for testing underlying assumptions about complex interventions and identifying effective components of interventions, tailoring interventions to patient groups and identifying subgroups in which treatment is likely to be successful [10]. Although relevant and meaningful qualitative studies exist, systematic reviews rarely include qualitative studies. Thus, the inclusion of qualitative studies in systematic reviews can help to explain discrepancies between the results of quantitative experimental and observational studies.

To overcome these limitations, a review approach was developed that goes beyond previous mixed-method review approaches. Pyramid Reviews [6] distinguish between experimental and observational research approaches as well as quantitative and qualitative approaches. Accordingly, quantitative-experimental, quantitative-observational, qualitative-experimental, and qualitative-observational partial reviews are conducted first—insofar as corresponding studies can be included—followed by a comparison or confrontation of the results from the different research approaches. Thus, all available evidence can be assessed, appropriate conclusions can be drawn and, if possible, recommendations for practice and research can be derived.

A Pyramid Review provides a foundation for gathering and synthesizing research evidence to answer clinical questions that inform everyday health care decisions, valuing and integrating external evidence from multiple research approaches. For each research approach, a distinction is made between three levels of evidence (LoE), which group together individual studies with comparable conclusiveness with regard to the effects investigated. In addition, there are systematic reviews, which, if done well, are more conclusive than the best individual studies, and expert opinions, which are the least conclusive form of external evidence. This results in the following assignment of levels of evidence and conclusiveness: LoE I (systematic reviews) = very high conclusiveness, LoE II = individual studies with high conclusiveness, LoE III = individual studies with medium conclusiveness, LoE IV = individual studies with low conclusiveness, and LoE V (expert opinions) = very low conclusiveness (see Figure 1). The initial assessment determines a preliminary LoE, which may change depending on the quality of the conduct and the quality of the results.

The inclusion of experimental and observational studies in a review can provide a more complete picture of the available evidence. According to that differentiation, statements can be made about the efficacy of an intervention under ideal conditions as well as the effectiveness under everyday conditions, and variations in effects can be analyzed. Moreover, the inclusion of qualitative studies allows for more complex, individualized, and meaningful analysis of the effects and, potentially, better explanations of statistical variance and contradictions.

The system and conduct of a Pyramid Review is based on the research pyramid and has been continuously developed since 2006 as a three-sided pyramid in the context of teaching research methodology for allied health professionals, particularly in Germany [6,12,13] and the US [14]. For the purpose of Pyramid Reviews, a four-sided variant was developed and published in German [6,13] and English [15] (see Figure 2).

This review aimed to evaluate the effects of home-based telerehabilitation interventions delivered to address upper extremity function in stroke patients. In our opinion, a Pyramid Review for clinical practice and research is currently the most meaningful form of evidence synthesis. It comprehensively considers evidence from different research approaches and evaluates it according to strict methodological standards. It also makes it clear which research approaches may be overrepresented and which are underrepresented in relation to a particular clinical question.

## 2. Materials and Methods

### 2.1. Pyramid Review: ABCDEF

A Pyramid Review is based on the usual steps for systematic reviews—but without restriction to specific research approaches or study designs. The reporting is based on the Preferred Reporting Items for Systematic Reviews and Meta-Analyses (PRISMA) guidelines [17]. The PRISMA checklist can be found in the supplements.

The following steps are specific to a Pyramid Review (see Figure 3):A.Assignment of the individual studies and reviews to the corresponding research approach on the research pyramidB.Preliminary appraisal of the level of evidenceC.Critical appraisal of the quality of the conduct of the study, and downgrading of the level of evidence, as indicatedD.Critical appraisal of the quality of the results, and upgrading of the level of evidence, if appropriateE.Clustering of the individual studies according to the research approach and conducting partial reviewsF.Critical synthesis of the overall state of evidence from the partial reviews

Steps B–E are carried out according to different procedures for quantitative and qualitative studies.

In order to conduct partial reviews of the results of the quantitative studies, separate meta-analyses are carried out for those studies that are clinically and methodologically similar enough (homogeneous studies). The majority of the quantitative studies included in this Pyramid Review are heterogeneous from a methodological point of view (RCT/non-RCT, comparative/non-comparative, heterogeneous control groups, chronic/subacute patients, different underlying technological concepts). Therefore, these heterogeneous studies are synthesized in a narrative review. With a narrative synthesis, heterogeneous studies can be better evaluated in subgroups in order to arrive at a comprehensible statement for these groups or individual treatment concepts.

In addition, for each study or outcome, the mean difference between baseline and postintervention was calculated and evaluated using the minimal detectable change (MCD) or the minimal clinically important difference (MCID) as well as Cohen’s d and the effect size r. The use of an outcome measure to assess change is an effective method for confirming an improvement in a patient’s health status resulting from an intervention. The assessment of change is based on a comparison of a measurable entity over a specified time period [18]. However, the determination of a clinically meaningful change in scores (MCID) serves as the benchmark for confirming a real change. The integration of statistical methods with a comprehensive clinical understanding can facilitate the identification of whether a change observed as a mere difference score is also a clinically meaningful one. This is the rationale behind our presentation of both the statistical significance and the MCID in this Pyramid Review.

For the qualitative evidence synthesis, the meta-aggregative approach was selected due to its unique ability to go beyond the mere generation of theoretical constructs and the methodologically differing qualitative research approaches [19] and to enable syntheses of results across the different methodological bases of qualitative research. The analytical strategy for the qualitative partial reviews employed in the meta-aggregative process comprises three stages: firstly, the extraction of findings from the primary studies; secondly, their further categorization; and thirdly, to synthesize findings in comprehensive sets.

This Pyramid Review also analyses three levels of plausibility, as delineated by Lockwood et al. [20]—“unequivocal”, “credible”, and “unsupported”—and does not include any unsupported statements. In the event that a study reports only unsupported results, it is excluded from the review in its entirety due to the presence of significant deficiencies in the quality of conduct.

The results of the quantitative and the findings of the qualitative analyses are based on a common set of concepts and categories, which allows for a direct correspondence between the sets and the primary and secondary outcomes. The categories are defined a priori in accordance with the research question, but inductive analysis could result in further categories.

The Pyramid Review is the critical synthesis of the overall state of evidence from the partial reviews. In terms of scientific rigor, the most robust and comprehensive evidence is that which is derived from studies that adhere to the highest standards of inference quality and inference transferability across all four research approaches [20].

### 2.2. Clinical Question

What effects do different approaches and technologies achieve in telerehabilitation for subacute and chronic stroke patients with motor impairments of the upper extremities (and how are these effects achieved)?

### 2.3. Eligibility Criteria

The following criteria were used to determine eligibility: (1) Sample consisted of individuals suffering from subacute (defined as <26 weeks) or chronic (defined as >26 weeks) stroke, aged 18 years or above; (2) Studies that assess the effectiveness or efficacy of synchronous telerehabilitation interventions (that is, technology-based distance therapeutic rehabilitation interventions delivered in real time under the supervision of a qualified therapist); (3) Studies that compare the efficacy of telerehabilitation interventions with a control group not using technology or no comparison; (4) The study outcomes were physical functioning, as defined by the International Classification of Functioning, Disability and Health (ICF) categories of body functioning, activities, and participation; (5) Any research approach, as defined by the research pyramid; (6) The languages were German, English, or French. We excluded studies that pertained to (1) repeat published studies; (2) conference proceedings and abstracts, study protocols, letters, discussions, or editorials; and (3) studies that provided incomplete data.

### 2.4. Search Strategy

To identify relevant studies, a search was conducted in PubMed, PubPsych, the Cochrane Central Register of Controlled Trials, SCOPUS, and CINAHL, from January 2000 to October 2024, on 24 October 2024. As both quantitative and qualitative studies were included in this Pyramid Review, the PICOS principle (participants, intervention, comparison, outcome, and study design) was employed to develop the search strategy for quantitative studies (see Figure 4) and the PerSPECTiF principle [21] (perspective, setting, phenomenon of interest/problem, environment, comparison, time, findings) was used for qualitative studies (see Figure 5).

The search terms used in this review included “stroke”, “cerebrovascular accident”, “vascular accident”, “telerehabilitation”, “telemedicine”, “telehealth”, “remote consultation”, and “teletherapy”. Additionally, the search terms included “upper and lower limb” and “dysphagia”. The retrieved records were limited to synchronous home-based physiotherapy, occupational therapy, and speech and language therapy. Furthermore, the reference lists of the original literature were manually searched for additional studies. The search strategies and specific details are presented in Appendix A.

### 2.5. Study Selection

All records identified in the search were imported into the reference management program CADIMA [22] for bibliography management and the elimination of duplicates. The screening process involved the initial examination of titles and abstracts, followed by a more comprehensive review of the full texts. The screening of titles, abstracts, and full texts were performed by two reviewers (K. S.-G., B.B.). In cases where the eligibility of a record was uncertain, it was discussed and resolved with a broader research group to reach a consensus.

### 2.6. Data Extraction

The data extraction from all studies was conducted using a self-designed and pilot-tested Excel spreadsheet that included the following items: study author, publication year, country, sample size, mean age, sex, time since stroke, type of stroke, lesion location, affected side, level of impairment at baseline, living arrangement, intervention and control methods, duration of intervention and follow-up, assessment tools and results, and profession of the treating therapist.

The data extraction from quantitative studies also includes the outcomes that measured physical functioning. The primary outcome was upper limb function, and the secondary outcomes were participation in daily activities, motivation, and adherence. The data extraction from qualitative studies was based on the meta-aggregative approach [23] and comprised findings that are related to the effects of the telerehabilitation interventions in general.

For the extraction of data from both quantitative and qualitative studies, a priori categories and concepts such as motor function, perceived effectiveness, motivation, and adherence were used in accordance with the research question of the review. In addition to this subsumptive approach, an inductive generation of categories related to effects on motor function was conducted.

In instances where the necessary data were not reported in the articles, the study authors were contacted via email. Data extraction was performed by one reviewer (K. S.-G.). Disagreements regarding the extracted results were discussed and resolved within the research team when necessary.

The present review was conducted in accordance with a registered protocol on the Open Science Framework (OSF) [24].

## 3. Results

### 3.1. Study Selection

A total of 747 records were identified in the search, and 596 records remained after duplicates were removed. Following preliminary selection based on a review of titles and abstracts, 470 studies were excluded from further consideration. In total, 126 studies were identified as potentially relevant and were assessed for eligibility. Of these, 62 studies were excluded following a full-text reading, and an additional 12 studies were excluded as they were reports of assessments administered in a digital format. As no systematic review identified through the literature search exclusively focused on synchronous telerehabilitation, they were excluded. Nevertheless, 77 eligible studies from these systematic reviews were added to the 51 already included publications in the Pyramid Review. In a subsequent step, 93 clinic-based studies were excluded, as well as one duplicate publication. A total of 32 studies met the eligibility criteria and were included in this Pyramid Review (Figure 6).

The present Pyramid Review is concerned with the evaluation of synchronous home-based studies on the upper extremity and includes 24 quantitative, four qualitative, and two mixed-methods studies with the experimental study approach. No observational studies could be found. The partial reviews are therefore limited to a quantitative-experimental and a qualitative-experimental partial review.

Of the 24 quantitative studies, nine were RCTs [25,26,27,28,29,30,31,32,33] and 15 were nonRCTs [34,35,36,37,38,39,40,41,42,43,44,45,46,47,48]. The data set of subjects presented by Piron et al. (2004) [41] is identical to that of the intervention group in the RCT published by the same authors in 2008 [28]. The quantitative data of the subjects published in the mixed-methods study by Standen et al. in 2015 [48] is identical with the intervention group of the RCT published in 2017 [30]. Furthermore, the data of the subjects published by Cramer et al. [36] and Podury et al. [42] is identical as well. All identical data were included only once in this review, resulting in 21 quantitative sub-studies, eight of which were conducted in Europe, 10 in North America, two in Asia, and one in New Zealand. In addition, six qualitative sub-studies [47,48,49,50,51,52] were undertaken, two of which were conducted in Europe and four in North America.

The approaches to telerehabilitation of motor deficits after stroke in the home environment analyzed in the studies are based on three different underlying technological concepts. Six studies were image-based interventions [32,33,34,44,46,47], which included the use of information and communication technologies (ICT) to facilitate remote communication between healthcare professionals and patients. These technologies, such as videoconferencing, are defined as “high-quality, interactive, bidirectional audiovisual systems” [53]. Sensor-based concepts were used in 12 studies [25,26,27,28,29,30,31,35,37,39,40,43], including three studies using functional electrical stimulation (FES) of the affected arm [29,35,37]. This conceptual group encompasses any type of robotic, gaming, and Kinect-based rehabilitation services that are delivered remotely. It includes terms such as “exergaming”, “serious games”, and “wearables”. The remaining two studies were based on virtual reality (VR) [36,37]. It is noteworthy that the sensor-based studies were also image-based, and that the VR-based studies employed sensors and a videoconferencing solution to facilitate the delivery of their intervention.

The evaluation and appraisal of the two experimental sub-studies (quantitative and qualitative) of the mixed-methods studies took place independently. The results of these sub-studies were incorporated into the overall results of the quantitative and qualitative sub-reviews.

The a priori defined categories and concepts are in reference to the effects that were extracted from the included studies. Figure 7 illustrates the categorization of the studies according to the subacute/chronic and severity of impairment.

### 3.2. Quantitative-Experimental Partial Reviews

#### 3.2.1. Participants

A total of 703 participants were identified across the 21 included quantitative sub-studies, with sample sizes ranging from 1 to 351. All participants were adults, comprising 351 therapists and 354 patients. In the included randomized controlled trials (RCTs), patients were randomly assigned to experimental groups (n = 121) and control groups (n = 114). The mean age of patients was 59.3 ± 6.3 years, with a range of 40.4 to 70 years. Nineteen sub-studies reported the sex of the patients, with a male prevalence of 63.9% ± 16.3. Nineteen studies described the time since stroke onset in the enrolled patients, with three reporting data from 80 subacute patients (<26 weeks from stroke onset) [32,33,36]. The mean age of the therapists is 40.4 ± 12.5 years, with a mean working experience of 14.8 ± 11.9 years. Of these therapists, 10% are male.

#### 3.2.2. Outcome Measures

A wide variety of outcome measurements were used to measure physical functioning of the upper extremity within the 21 quantitative studies (Figure 8). These measurements were used to examine the survivor of stroke’s upper extremity’s function pre and post intervention.

The Wolf Motor Function Test (WMFT) was the most popular among the outcome measures evaluating activity according to the ICF framework, followed by the Nine Hole Peg Test (NHPT) and the Action Research Arm Test (ARAT), the Box and Block Test (BBT), and the Chedoke Arm and Hand Activity Inventory (CAHAI). As illustrated in Figure 7, the Fugl–Meyer Assessment (FMA) for measuring function was the most utilized outcome measure. In order to assess participation, the Motor Activity Log (MAL) and the Stroke Impact Scale (SIS) were employed.

Among the included tools are assessments scored by healthcare professionals, such as the FMA-UE and BBT, as well as questionnaires completed by study participants, including the MAL and SIS.

The WMFT [54] is an assessment tool designed to evaluate the ability to use the upper limb after a stroke or traumatic brain injury, encompassing both simple and complex movements as well as functional activities. The functional tasks are organized in a hierarchical structure, progressing from simple to complex and from proximal to distal. The WMFT is comprised of 15 function-based tasks, which are divided into two categories: Performance time (WMFT-TIME) and Functional ability (WMFT-FAS). Additionally, it incorporates two strength-based tasks and employs a six-point ordinal scale, ranging from “0” (indicating a lack of attempt with the involved arm) to “5” (demonstrating arm participation and normal movement). The maximum score is 75, with lower scores reflecting lower functioning levels. The performance of the affected upper limb is assessed by reference to the opposite side. The distinctive aspect of this assessment is its evaluation based on a video recording, in consideration of the criteria “quality of movement” and “time required to perform”.The NHPT/9HPT [55] is used to measure finger dexterity in patients with various neurological diagnoses. The test is conducted by requesting that the patient remove the pegs from a container and insert them into the holes on the board in the most time-efficient manner possible. Subsequently, the patient must then remove the pegs from the holes, one by one, and replace them back into the container. The time taken to complete the test activity is then recorded in seconds and used to score the patient.It should be used in association with other upper extremity performance tests in order to estimate upper limb function with more accuracy. Additionally, the NHPT can be conducted relatively inexpensively and is quick to administer.The ARAT [56] is a tool for recording unilateral arm function at the activity level, for objectively assessing movement performance, and for even making predictions. The 19-item measure is divided into four sub-tests (grasp, grip, pinch, and gross arm movement) designed to assess the functional performance of the upper extremity through observation. The degree of success achieved in performing each item is evaluated on a four-point ordinal rating scale, with values ranging from 3 to 0.(3) Performs test normally(2) The subject completes the test, albeit at a slower pace or with evident difficulty.(1) The subject performs the test partially.(0) The subject is unable to perform any part of the test.The maximum score on the ARAT is 57 points (possible range 0 to 57). A higher score indicates greater functioning in the upper extremity. In addition to clinical application, it is also used internationally in (efficacy) studies.The BBT [57] is a tool used to assess unilateral gross manual dexterity. The patient is seated at a table, facing a rectangular box that is divided into two square compartments of equal dimension by means of a partition. One hundred and fifty wooden blocks, measuring 2.5 cm, are placed in one compartment or the other. The patient is instructed to move as many blocks as possible, one at a time, from one compartment to the other for a period of 60 s. The BBT is scored by counting the number of blocks carried over the partition from one compartment to the other during the 1-min trial period. Higher scores on the test indicate better gross manual dexterity. The BBT requires between 2 and 5 min to complete, making it a relatively quick assessment to administer.The ABILHAND [58] is a measure of manual ability for adults with upper limb impairments. The scale assesses manual ability, defined as a person’s capacity to manage daily activities that require the use of the upper limbs. It is focused on the patient’s perceived difficulty. The assessment for chronic stroke patients of 23 items of bimanual activities employs a three-level response scale for each item, with the following rating options: “Impossible” = 0, “Difficult” = 1, “Easy” = 2, and “Not applicable” = missing data.The CAHAI-7 [59] is an upper extremity measure that uses a seven-point quantitative scale to assess functional recovery of the arm and hand after stroke. Patients are asked to complete the items bilaterally. A score of 1 = patient requires total assistance, and the weak upper limb performs less than 25% of the task. A score of 2 = patient requires maximum assistance, and the weak upper limb performs 25% to 49% of the task. There are no signs of arm or hand manipulation, only stabilization. A score of 3 = client requires moderate assistance, and the weak upper extremity performs 50% to 74% of the task. Begins to show signs of arm or hand manipulation. A score of 4 = client requires minimal assistance (light touch), and the weak upper limb performs more than 75% of the task. A score of 5 = client requires supervision, coaxing, or cueing. A score of 6 = client requires the use of assistive devices or requires more than reasonable time, or there are safety concerns. And a score of 7 = completely independent in performing the task. The minimal possible score for the CAHAI is 7 and the maximum is 49, with higher scores indicating greater functional independence The purpose of this measure is to assess the functional ability of the paretic arm and hand to perform tasks that stroke patients have identified as important after stroke. Administration time for the seven-item version is approximately 15 min.The FMA [60] is a stroke-specific measuring instrument designed for the evaluation of sensorimotor functions and balance, joint mobility, and joint pain. The assessment comprises a series of unilateral tasks and movement tasks. It is also possible to utilize single subscales; for example, the one pertaining to upper limb motor function (FMA-UE) independently. The maximum score for the FMA-UE is 66. Items are scored on a three-point ordinal scale, with 0 indicating that the task cannot be performed, 1 indicating that the task is performed partially, and 2 indicating that the task is performed fully. The FMA is employed in both clinical practice and research settings. However, the test necessitates sustained patient attention, which constrains its applicability in individuals with attention deficit disorders.The MAL [61] is a diagnostic tool comprising a structured interview that assesses the use of the affected upper limb in everyday life following a stroke. In the interview, the patient and/or their relative should assess the amount of use (AOU) and the quality of use (QOU) of the affected upper limb. The MAL is a standardized instrument, which increases its validity and reproducibility in both everyday clinical practice and research. The MAL is not suitable for the assessment of patients with severe limitations of the upper extremity or cognitive impairments.The Stroke Impact Scale (SIS) [62] is used to evaluate disability and quality of life after a stroke. The SIS-16 has 16 items on strength, hand function, mobility, and ADL/IADL. Patients rate the difficulty of each item over 2 weeks on a five-point Likert scale.1 = could not do it at all2 = very difficult3 = somewhat difficult4 = a little difficult5 = not difficult at all

The SIS may be completed by the patient or an appropriate proxy. A higher score indicates a higher quality of life and level of participation.

The period following a stroke is frequently divided into distinct phases. The Stroke Roundtable Consortium has proposed a classification system wherein the initial 24 h are designated as the hyperacute phase, the first 7 days as the acute phase, the initial 6 months as the subacute phase, and the period extending from 6 months onwards as the chronic phase [63]. This Pyramid Review examined the findings of the primary studies on two patient groups: those with subacute conditions and those with chronic illnesses. Details and characteristics can be found in Appendix A.

### 3.3. Subacute Studies

Table 1 reports data from subacute patients (<26 weeks post-stroke). The RCT by Wu et al. [32] is exclusively focused on patients with subacute stroke, whereas the studies by Bernocchi et al. [34] and Cramer et al. [36] permit a distinct examination of data from patients whose stroke occurred less than or more than 26 weeks ago. The studies by Wu et al. and Bernocchi et al. are image-based trials, whereas the study by Cramer et al. [36] is a VR-based trial. Both image-based studies represent a shift from the clinic-based model, in which healthcare practitioners can closely monitor their patients and provide support to them and their families as a team, to telerehabilitation. This includes the involvement of a whole healthcare team that addresses the needs of both patients and caregivers.

With regard to the findings of Cramer et al., a moderate improvement in motor function is evident, with the baseline value of 46.90 points increasing to a post-intervention value of 56.50 measured by the FMA-UE. This represents a mean difference of 9.60 points. Wu et al. demonstrate a more pronounced improvement, commencing from a lower baseline of 11.93 points and attaining a post-intervention value of 55.33, which corresponds to a mean difference of 43.40 points. Both studies demonstrate a positive effect for the interventions, with Wu et al. showing a notably larger improvement compared to Cramer et al. [36].

The study conducted by Cramer et al. [36] revealed a percent change of 20.47% (Cohen’s d: 0.19). The percent change of 20.47% indicates a moderate degree of improvement. Cohen’s d of 0.19 indicates a small effect size. The mean difference of 9.60 points is below the minimal clinically important difference (MCID) for moderate subacute patient cases, which is identified as 12.40 points for the FMA-UE [64].

In contrast, the study of Wu et al. demonstrates a percent change of 363.79% (Cohen’s d: 1.29), which suggests a very large improvement. This is supported by the observation that Cohen’s d of 1.29 indicates a very large effect size, which is well above the threshold for a large effect (0.8). The mean difference of 43.40 points observed in the intervention group exceeds the minimal clinically important difference (MCID) for moderate to severe subacute patient populations, which has been identified as 12.40 points for the FMA-UE [64].

The analysis for dexterity measured by the Box and Block Test results (secondary outcome) from the study conducted by Cramer et al. [36] indicates a notable enhancement in the scores following 12 weeks of telerehabilitation. The mean score increased from 31.5 at the baseline measurement to 45.4 following the intervention. The *p*-value of 0.0005 indicates that this improvement is highly statistically significant. The value of Cohen’s d, at 4.63, suggests a very large effect size (r = 0.85). This suggests that the telerehabilitation intervention had a notable and meaningful impact on the subjects’ hand dexterity. The standard deviation exhibited a notable decline, from 4.1 at the baseline to 1.9 after 12 weeks. This suggests that the subjects’ performances became more consistent following the intervention. Moreover, the mean difference of 13.9 is considerably above the minimum detectable change (MDC), representing an improvement of 5.5 blocks per minute [65].

In their study, Bernocchi et al. used the NHPT to measure finger dexterity, also known as fine manual dexterity. From a baseline time of 50.2 s, subjects improved to 30.4 s, a significant improvement of 19.8 s (*p* = 0.01). The value of Cohen’s d, at 1.22, and the effect size r (0.52) indicate a large effect. This implies that the telerehabilitation intervention had a notable and meaningful impact on the subjects’ finger dexterity. However, the identified MCID of 32.8 s [65] is significantly higher than the reached mean difference of 19.8 s.

In the studies investigating the responsiveness and validity of dexterous function measures in stroke rehabilitation the BBT and NHPT demonstrated adequate to excellent correlations at both the pre-treatment (ranging from rho = −0.55 to −0.80) and post-treatment (ranging from rho = −0.57 to −0.71) stages [66]. On this basis, it can be assumed that the two test results are directly comparable, and it can be concluded that the treatment itself or the dose of the treatment conducted by Cramer et al. [36] had a greater effect than the treatment administered by Bernocchi et al.

### 3.4. Chronic Studies

#### 3.4.1. Image-Based Studies

This subsection presents the results of image-based interventions in chronic stroke patients (see Table 2). As previously noted, Bernocchi et al. published data on both subacute and chronic participants, while Yang et al. focused exclusively on chronic participants [34,56]. Both studies demonstrate a successful and evidence-based approach to digital implementation. While Bernocchi et al. [34] adopted a complex approach from inpatient care for stroke patients and their caregivers, Yang et al. [46] transferred the Graded Repetitive Arm Supplementary Program (GRASP) [67], which is a standard practice in Canada.

As can be seen from Table 2, for measuring the effects of the treatments both authors utilized disparate outcome measures. Yang et al. developed a novel outcome measure, the Arm Capacity Measure (ArmCAM), which facilitates the remote evaluation of upper extremity motor function following stroke. Although the *p*-value reported in the study (*p* = 0.016) suggests statistical significance, the effect sizes (Cohen’s d: 0.16 and r: 0.08) indicate that the practical significance of the change is relatively small. The results for hand function, as measured as secondary outcome by the SIS, demonstrate a statistically significant increase (*p* = 0.03) and yield a moderate effect size (Cohen’s d: 0.56; effect size r: 0.27), indicating a meaningful improvement in hand function following treatment. Conversely, the mean difference pre-post (12 points) did not achieve the MCID for chronic stroke patients, which is 17.8 points [68].

Bernocchi assessed finger dexterity with the Nine Hole Peg Test (NHPT). A reduction in time pre-post of 9 s suggests a moderate effect (Cohen’s d: 0.64; effect size r: 0.3054), yet it is considerably less than the published MCID of −32.8 s [69].

#### 3.4.2. Sensor-Based Studies for Chronic Stroke Patients

Functional electrical stimulation (FES) is a technique utilized to replace or facilitate a muscle contraction during a specific functional activity by applying an electrical current to the nerves that are responsible for controlling muscles. The objective of this treatment modality is to enhance muscle contraction and improve motor control.

As can be seen in Table 3, Hermann et al. [37] and Prathum et al. [29] utilized the FMA-UE as a primary outcome to measure motor function whereby it is important to note that the Hermann study is a single-case study, which is critical for a broader comparison.

The single patient in the Hermann [37] study had more severe impairments at baseline and started with a lower baseline (25) compared to Prathum (IG) (36.33) [29].

After 3 weeks, the absolute improvement was two points, which is below the MCID for severely impaired chronic patients identified at 3.5 points [56]. The relative improvement was 8%. Patients in the intervention group of the RCT conducted by Prathum et al. [29] were less impaired and showed a greater absolute mean improvement of 7.17 points, which is within the range identified for the MCID of minimally to moderately impaired patients with chronic stroke of +4.25 to +7.25 points [70]. The relative improvement is 19.7%.

In conclusion, while both studies show improvement after 3 weeks of treatment, the Prathum (IG) study shows a greater treatment effect in both absolute and relative terms [29]. This may indicate that the intervention was more effective in the Prathum (IG) study because the application of 20 min of dual tDCS contributed to the greater improvement [29].

Buick et al. utilized the ARAT as a primary outcome to measure hand function, whereas Hermann et al. used it a secondary outcome [35,37]. As mentioned above, the Hermann study is a single-case study (n = 1), which significantly limits its generalizability and statistical power [37]. The Buick study, with 11 participants, provides more reliable and potentially generalizable results due to the larger sample size [35].

The Hermann [37] study started with a lower baseline (10.0) compared to the Buick study [37] (16.7). This difference in baseline makes direct comparisons of absolute improvement difficult.

The Buick [35] study showed a significant improvement of 5.8 points (Cohen’s d: 5.8; effect size r: 0.95), which is above the MCID for chronic stroke patients, which is +5.7 points [71]. The Hermann study [37] showed a greater improvement of 8.0 points, an increase of 47.9% from baseline. As it is a single-case study, no effect sizes could be calculated.

The Buick study [35] provides good evidence of treatment efficacy due to the remarkable improvement in ARAT scores, and it has a larger sample size.

#### 3.4.3. Digital CIMT Studies

Constraint-induced movement therapy (CIMT), developed by Edward Taub in 1994 [72], is a rehabilitation approach that primarily aims to improve motor function in individuals with significant upper extremity impairments resulting from a stroke. The three fundamental principles of CIMT are the principles of constraint, massed practice, and shaping. Constraint-induced movement therapy (CIMT) entails the use of a mitt to constrain the non-affected limb during the daytime, thereby encouraging the use of the affected limb. This compels the patient to rely on and practice movements with the impaired limb. The therapy places an emphasis on the repetition and intensity of practice with the affected limb. It incorporates a “shaping” technique, wherein smaller, achievable goals are set, with the difficulty level increasing as the patient gains proficiency. This helps to build confidence and encourages continuous engagement in everyday tasks. CIMT, in its standard format, requires 35 h of individual, face-to-face therapy, typically conducted by a qualified physical or occupational therapist. The protocols of the three included studies [31,44,47] were adapted from the original design, focusing on maintaining the three core components of CIMT, including intensive and graded use of the affected limb, constraining the unaffected limb, and the transfer package, which involves using the limb in daily activities (see Table 4). The modified CIMT (mCIMT) protocol comprises a series of treatments, with modifications to the restraint procedures and an extended distribution of therapy over a longer period. Page et al. [47], Smith et al. [44], and Saygili et al. [33] used a more recent approach with an image-based constraint-induced movement therapy (iCIMT) approach, which is a modified constraint-induced movement therapy approach that incorporates telehealth. Smith et al. [44] conducted a study comprising two distinct patient groups: one exhibiting “high function”, characterized by mild to moderate impairment, and another displaying “low function”, defined by moderate to severe impairment. These groups were evaluated separately. Uswatte et al. [31] used an automated, upper extremity workstation with integrated sensors and video cameras that was installed in the homes of participants. Internet-based audiovisual and data links enabled remote supervision of the treatment by a therapist monitoring the session from a location in the laboratory.

All four studies utilized the MAL as primary outcome. The CIMT treatment shows varying effects for the amount of use (AOU) across the two studies with moderate to severely impaired patients. The study conducted by Page (2007) [47] illustrates a clinically significant enhancement in the amount of use (AOU) of the affected hand, with an increase of 2.15 points. The Smith (2020) [44] study demonstrates improvement, with a score change of +0.73, but this falls short of meeting the established MCID threshold of +0.89 for a clinical difference [73].

Two distinct meta-analyses were performed, one for patients exhibiting moderate-severe symptoms (encompassing Smith’s “lower function” and Page et al.) and another for those with mild-moderate symptoms (including Smith’s “higher function”, Uswatte et al., and Saygili et al. [33]).

One meta-analysis was conducted to assess the mean difference in MAL (amount of use) of the Page and Smith (lower function) values. As can be seen in Figure 9, the weighted mean difference was 1.74, with a 95% confidence interval of [1.55, 1.92] and a Z-score of 18.33. The *p*-value was 0.0000, and the I-squared was 97.8%. Furthermore, the results for the quality of use (QOU) also indicate that the Page (2007) [47] study demonstrates a clinically significant improvement of +2.0 points, whereas the Smith (2020) [44] study shows a less pronounced improvement of +0.69, which does not meet the MCID threshold of +0.77.

The results of the three studies, which treated mild to moderately affected chronic stroke patients, indicate that the Uswatte (2021) [34] intervention group exhibited a clinically significant improvement of +2.4 points for the AOU (Cohen’s d: 12.5; effect size r: 0.99), whereas the subgroup of the Smith (2020) [44] study with higher function demonstrated a less pronounced improvement of +0.79 (Cohen’s d: 0.782; effect size r: 0.365), which did not meet the MCID threshold of +0.89.

A second meta-analysis was conducted on the MAL (amount of use) of the Uswatte, Saygili, and Smith (higher function) studies [33,34,44]. The weighted mean difference was 2.41 (95% CI: [2.24, 2.58]), with a high Z-score of 135.05 and a *p*-value of 0.0000 indicating substantial heterogeneity among studies, suggesting that almost all of the observed variance is due to real differences among studies rather than chance. The I-squared value was 99.8%. As can be seen in Figure 10, the weighted mean difference of 2.41 indicates a substantial improvement in the amount of arm use (AOU), exceeding the MCID of +0.89 and suggesting a clinically significant effect.

With regard to the findings from the WMFT, all studies yielded statistically significant results that exceeded the established threshold for the minimal clinically important differences (MCID) for time and function [74].

It is crucial to acknowledge that additional variables, such as the sample size (n = 4 for Page vs. n = 13 and n = 15 for Smith, n = 10 for Uswatte (IG) and Saygili (IG), respectively [33,34,44]) and disparate treatment protocols, may potentially contribute to the discrepancy in outcomes among the four studies. Page administered 15 h of individual therapy over 10 weeks, whereas Smith [44] provided a combination of individual therapy and group sessions, resulting in 21 h of treatment in 6 weeks, Uswatte administered 35 h of treatment in 2 weeks and Saygili 22.5 h of treatment in 3 weeks [33,34].

#### 3.4.4. Other Sensor-Based Studies

The data from the Uswatte et al. study [34], which was conducted using a sensor-based approach, has already been extracted and evaluated in the context of CIMT studies. The remaining eight studies were divided into two groups, as four of them were published by the same authors (Piron et al. [26,27,28,40,41]) and utilized the same sensor-based 3D motion tracking device.

The studies conducted by Piron with the VRRS.net consistently yielded positive results for the included mild-moderate impaired patients (see Table 5). Two of the three randomized controlled trials (RCTs) demonstrated statistically significant results in the pre-post comparison for the intervention groups (2008 + 2009).

Three of the four Piron studies exceed the minimal clinically important difference (MCID) of 4.25 points [70], as indicated by their mean score difference. The fourth study exhibits a mean difference of 4, which is slightly below the aforementioned value. To calculate the effect size of these sensor-based interventions, we employed the formula for Cohen’s d for repeated measures, which accounts for the correlation between pre- and post-intervention measurements. In the absence of the actual correlation, we assumed a moderate correlation of 0.5, which is a common practice when the true correlation is unknown. All studies demonstrate medium to large effect sizes, with the non-RCT (2002) exhibiting the most pronounced effect (Cohen’s d = 0.814) and the intervention group of Piron (2009) [27] demonstrating the smallest (though still medium-sized) effect (Cohen’s d = 0.609). The Cohen’s d values for the intervention groups of Piron (2006 and 2008) [26,28] were 0.629 and 0.671, respectively. The results indicate that the interventions had a substantial positive impact on FMA-UE scores across all studies.

The meta-analysis of the results indicates a statistically significant overall effect of the interventions across the Piron studies, with a weighted mean difference of 4.70 and a 95% confidence interval of [2.67, 6.73]. The *p*-value is effectively zero, suggesting strong evidence against the null hypothesis of no effect. The I-squared value of 0.0% indicates no observed heterogeneity among the studies, meaning the results are consistent across the different studies. The forest plot (Figure 11) visually supports these findings, showing that the pooled effect exceeds the Minimal Clinically Important Difference (MCID) of 4.25, suggesting that the interventions have a clinically meaningful impact.

No statements can be made about the results obtained by Piron et al. 2009 [27] with the ABILHAND assessment for secondary outcome, as no primary data are available.

### 3.5. Exergaming (Sensor-Based)

The remaining four sensor-based studies (see Table 6 below) concentrated on exergaming, a neologism derived from the terms “exercise” and “gaming”. The studies employed three-dimensional motion capture techniques to analyze hand and arm movements for moderate-severe affected patients.

The studies of Allegue et al. [25] and Jordan et al. [39] used also the FMA-UE as primary outcome measure. While the patients in the Allegue study [25] exhibited no change in function (mean difference = 0 points), the mean difference in the Jordan study [39] patients was 4.8, which is well above the MCID for moderate-to-severe affected chronic stroke patients of 3.5 points [56] showing a huge effect (Cohen’s d = 1.23; effect size r = 0.52).

In their respective studies, Standen et al. [30] and Allegue et al. [25] employed the MAL as a secondary outcome to quantify the amount of use (AOU) of the affected hand. The results of both studies fell below the MCID for chronic stroke (moderate to severe impairment): + 0.89 [73]. The mean difference of the patients included in the Allegue study [25] was 0.6 for the AOU and QOM, respectively, while the effect measured was high (Cohen’s d = 0.81; r = 0.38 (AOU); Cohen’s d = 1.09; r = 0.48 (QOM). In the patients included in the Standen study [30], the mean difference was 0.4 for the AOU and showing a low effect (Cohen’s d = 0.13; r = 0.07). The minimum clinically important difference (MCID) for quality of movement (QOM) was found to be +0.77, with the mean difference for this in patients of the Standen study [30] being 0.83 showing a medium effect (Cohen’s d = 0.32; r = 0.16).

Allegue et al. [25] used the SIS as secondary outcome for assessing quality of life and level of participation. The results indicate small effect sizes for all measures, with hand function (Cohen’s d: −0.1453; Effect size r: 0.0725) and ADL (Cohen’s d: −0.1086; Effect size r: 0.0542) showing negative Cohen’s d values because of a decrease in scores post-treatment (hand function: −6.25; ADL: −2.5), while mobility shows a slight increase (Cohen’s d: 0.1559; Effect size r: 0.0777). The loss/gain post-treatment across all areas was considerably less than the published minimal clinically important difference (MCID) values for chronic stroke patients [66,68].

The single-case study by Reinkensmeyer et al. [43] is the only study to employ the CAHAI 7 assessment. The patient exhibited an increase in hand motor function from one point prior to treatment to two points post-treatment. However, this represents a limited improvement, given that the MCID for severely affected chronic stroke patients is 6.3 points [69].

Sobrepera et al. [45] assessed the potential usefulness of a socially assistive robot (SAR) designed for telerehabilitation. The objective of the SAR is to enhance the efficacy of telerehabilitation by facilitating better communication, augmenting patient motivation, and improving compliance during therapy sessions. The robot features a humanoid interface for patient interaction, remote operation capabilities for clinicians, two movable upper limbs for demonstrating exercises, speech capabilities for communication and instruction, and a computer vision system for automated assessments.

A total of 351 therapists were surveyed to gain insight into their perceptions of the potential barriers and facilitators associated with the use of a SAR. On a scale of 0 to 100, with 0 indicating a lack of interest and 100 indicating a high level of interest, 63.8% of respondents indicated a rating of less than 50, and 27.1% indicated a rating of 50 or above. The therapists who expressed interest in utilizing Lil’Flo perceived its usefulness to be significantly higher than those who did not.

### 3.6. VR-Based Studies

Virtual reality (VR) systems integrate a range of neuroscientific and motor learning theories. The games facilitate task-oriented practice and repetitive, modifiable difficulty, thereby enabling the delivery of individualized therapy. The two studies (see Table 7) that are the subject of this evaluation employ a non-immersive VR-based approach.

The effect size observed in both studies for the primary outcome FMA-UE is slightly above 0.5. Cramer (2021) [36] reported a value of d = 0.55, while the mean difference was 4.8.

Holden (2007) [38] reported a standardized effect size of d = 0.59, with a mean difference of: 7.50.

Both studies exceed the MCID for FMA-UE in chronic stroke patients with mild-moderate impairment, with effect sizes ranging from +4.25 to +7.25 points.

The Holden (2007) [38] study shows a slightly larger effect, but both are in a comparable range. The Cramer study [36] had a smaller sample size than Holden et al. [38], which may impact the reliability of the effect size estimate. The similarity between the two effect sizes indicates that both studies identified comparable levels of improvement. These effect sizes indicate that the interventions resulted in meaningful improvements for patients. The changes are clinically relevant and exceed the identified MCID for the FMA-UE in chronic stroke patients.

The effect size for the secondary outcome (BBT) utilized by Cramer et al. [36] was found to be Cohen’s d = 0.97; r = 0.40. This resulted in a gain of 4.8 blocks, which is below the identified MCID of 5.5 [65].

The effect sizes for the secondary outcome (WMFT) utilized by Holden et al. [38] were found to be Cohen’s d = 0.5 for time, d = 0.49 for grip strength and d = 0.66 for hand to box/shoulder strength, respectively, and the mean difference in time (−15.6 s) was better than the published MCID of −1.5 to −2 s [74].

Details of the quality of conduct and the quality of results can be found in the Appendix A.

### 3.7. Qualitative-Experimental Partial-Reviews

#### 3.7.1. Participants

The six qualitative sub-studies focused on three key groups: patients (n = 55), therapists (n = 12), and family members of patients (n = 9). A total of five studies included patients with a mean age of 61.51 (10.75) years, with the male to female ratio being 73% to 27%, respectively. Only one study conducted interviews with subacute patients [49], while Donnelly et al. [50], Page et al. [47], Sivan et al. [52], and Standen et al. [30] employed semi-structured interviews with chronic stroke patients with a mean time since stroke of approximately 4 years (48.14 months). However, there is considerable variability in the data, as evidenced by the high pooled standard deviation of 51.18 months. This indicates that the time since stroke varies considerably among the patients in these studies, spanning from relatively recent strokes to those that occurred several years ago.

#### 3.7.2. Data Extraction and Categorization

Table 8 illustrates the diverse range of qualitative methodologies employed in the study. In addition to three studies that conducted semi-structured interviews, one study used focus group interviews and another employed informal interviews. Four studies employed thematic analysis for data analysis, one utilized content analysis based on the core set for stroke (ICF), and one did not publish the analytical approach used for data analysis.

In light of the preliminary appraisal of the individual studies (Step B of the Pyramid Review) included in our analysis, we proceeded to evaluate the quality of conduct (Step C). In order to accomplish this, we drew upon the framework developed by Teesside University for the critical appraisal of qualitative studies [75]. The details of the results of this appraisal can be found in Appendix A.

Table 9 illustrates the level of evidence in accordance with the systematic approach of the Pyramid Review, categorized as Step B, C, and D.

### 3.8. Aggregation of Findings

A total of five studies were deemed suitable for inclusion in the meta-aggregation. The qualitative sub-study of the mixed-methods trial by Page et al. was downgraded from LoE IV to LoE V due to the low quality of its conduct. According to the meta-aggregation approach, it was excluded because it published only unsupported findings.

Following the same rationale as the quantitative studies, the findings were analyzed separately for patients with subacute and chronic conditions. Chen et al. [49], employing an image-based approach, addressed the perceptions and experiences of subacute patients and their therapists. Four additional studies employing sensor-based approaches provided insight into the findings from the perspective of chronic patients, their therapists, and caregivers.

### 3.9. Subacute Studies (Image-Based Approach)

The findings from the study by Chen et al. regarding the effects and improvements in motor function among subacute stroke patients are as follows:Enhanced Physical Abilities: Some participants reported improvements in dexterity, strength, and endurance. They noted a significant difference in how their arms functioned at the end of therapy compared to before therapy, indicating tangible physical improvements. The potential factors responsible for the observed improvement were not identified.Therapist Interaction: During therapy sessions, therapists engaged with patients through various games and exercises. They monitored participant movements and provided verbal corrections, adjustments, and answers to questions. From the perspective of the therapists, this interactive approach was associated with the observed improvements in physical functioning.

Overall, the study suggests that structured therapy sessions, combined with active therapist involvement, lead to enhanced physical abilities in subacute stroke patients.

The findings from the study by Chen et al. indicate several key points related to the perceived improvement in motivation and adherence among subacute stroke patients:Convenience and Accessibility: The convenience of having therapy sessions at a location and time that suits the patient led to higher doses of therapy. This suggests that removing the need to travel to a therapist at a scheduled time can enhance therapy adherence.Social Support: Although the system was designed for individual use, participants noted that receiving attention and support from friends and family was a significant motivator. This highlights the importance of social support in maintaining engagement with therapy.Future Use: Most participants expressed a desire to continue using the system in the future, indicating a positive reception and perceived benefit from the therapy system as well as having fun while playing the exergames.

Overall, these findings suggest that convenience, social support, and positive user experience contribute to improved motivation and (future) adherence to therapy among subacute stroke patients.

### 3.10. Chronic Studies (Sensor-Based Approach)

The findings of the studies indicated that patients who underwent sensor-based telerehabilitation experienced an improvement in their physical functioning. Specifically, they reported an enhancement in their arm strength and mobility, a greater reliance on the affected arm for daily activities, and an overall improvement in their functional abilities. They also mentioned a high level of satisfaction with this approach. Key reasons mentioned by the patients for this satisfaction included the enjoyable and stimulating experience of using computer-based therapy and the ability to perform rehabilitation at home. While some patients felt that the assistance provided was insufficient for their abilities, all cited participants appreciated the game-based therapy concept. The use of visual biofeedback further enhanced movement attempts.

Therapists expressed interest in the system’s ability to track individual performance and provide feedback during therapy. In addition, both patients and therapists believed in the therapeutic value of the program, with therapists advocating for functional exercises and specific tools to be incorporated into training programs.

Furthermore, the caregivers indicated a high level of satisfaction with the perceived effects of the sensor-based approach on their family members.

The findings on perceived improvements in motivation and adherence among chronic stroke patients treated with sensor-based telerehabilitation highlight several key points:Feedback and meaningful use: Patients were motivated and satisfied by receiving feedback and found it meaningful to actively use their affected arm.Gamification: The use of games in rehabilitation increased motivation and acceptance. The competitive and immersive nature of the games contributed to engagement, although some patients with prior gaming experience found the games boring over time.Score tracking: Displaying scores after each session helped patients track their progress and motivated them to improve their performance.Social gamification: Although the system was single-player, social factors such as community and peer approval played a motivational role for patients.Study participation: Some patients were motivated by their participation in the research study itself.

Overall, the combination of feedback, gamification, and performance tracking had a positive effect on patient motivation and adherence to the therapy program.

The capacity of the sensor-based approaches to automatically adjust the level of assistance based on individual performance was highlighted as a valuable feature, as it enabled therapists to maintain high levels of motivation and adherence among their patients.

## 4. Discussion

The utilization of digital health interventions (DHI) [76] in the treatment of stroke patients is becoming increasingly prevalent in neurorehabilitation. Nevertheless, it remains uncertain whether synchronous home-based telerehabilitation is an effective means of supporting upper limb recovery. In order to gain insight into this matter, we conducted a Pyramid Review, with the objective of developing a comprehensive and nuanced understanding of the effectiveness and efficacy of telerehabilitation for these patients.

A Pyramid Review is a relatively new review approach developed predominantly in Germany. There are currently only two English-language publications on the research pyramid [14,15], which has been and continues to be perceived and cited internationally [77,78].

We found this approach useful as there was a considerable diversity across the included studies with respect to the research approaches, the eligibility criteria, interventions, underlying technologies for telerehabilitation, and outcome measures.

For us, evidence from this approach is considered to be the best possible for informing scientific and practical knowledge. It is characterized by the differentiation and confirmation of the efficacy and effectiveness of interventions that are adequately described and corresponds to the highest current standards of methodology for each research approach. This encompasses both the appropriateness of the study types or designs employed and the quality of implementation.

Regarding the efficacy for motor function, the results suggest that telerehabilitation interventions can lead to improvements in upper extremity function for both subacute and chronic stroke patients, though the magnitude of effects varied across studies.

### 4.1. Effects in Subacute Patients

For subacute patients, image-based and VR-based approaches showed promising results. Wu et al.’s image-based study demonstrated large improvements exceeding minimal clinically important difference (MCID) thresholds on the Fugl–Meyer Assessment Upper Extremity (FMA-UE) scale. The VR-based intervention by Cramer et al. [36] also yielded positive effects, though more modest in magnitude.

The studies conducted by Bernocchi et al. and Wu et al. indicate that a collaborative care approach is beneficial for patients with subacute conditions. This connection is also identified by the therapists in the interviews and confirmed in the questionnaires of the respective quantitative studies on Stroke-Specific Quality of Life and the Family Strain Questionnaire. It appears that telerehabilitation for subacute stroke patients necessitates structured therapy sessions and therapist involvement to facilitate physical improvements in these patients. Additionally, convenience, social support, and a positive user experience are essential to enhance motivation and adherence during the subacute phase after stroke. These elements are vital to comprehending the overall efficacy of the therapy for subacute stroke patients.

### 4.2. Effects in Chronic Patients

Nevertheless, the same studies did not yield evidence of an equally great effect for chronic patients. Although patients did continue to benefit from the treatment, the absence of qualitative findings that could facilitate a deeper understanding of the phenomenon and reasons for a lower effect are lacking insights into the diverse perspectives.

Constraint-induced movement therapy (CIMT) adaptations consistently produced clinically meaningful improvements across multiple studies. However, the case numbers of the two subgroups (mild-moderate and moderate-severe) are so small that the statements about the sometimes very high effects achieved cannot be generalized. The only qualitative sub-study (mixed methods) that could have reported a deeper understanding of interdependencies was excluded from the meta-aggregation due to major weaknesses in the quality of the implementation and results. The remaining qualitative studies providing evidence on sensor-based telerehabilitation in chronic patients are exergaming studies.

The sensor-based interventions by Piron et al. demonstrated medium to large effect sizes on the FMA-UE but have been downgraded one level of evidence, respectively, due to weaknesses in the quality of conduct.

VR-based interventions by Cramer and Holden showed improvements exceeding MCID thresholds, whereby the effects of the secondary outcome in particular were extraordinarily high in Cramer et al. [36,38].

However, results were mixed for some exergaming approaches, with some studies showing minimal or no improvements. This highlights the importance of intervention design and implementation.

### 4.3. User Experiences and Adherence

The results of the quantitative studies and the findings of the qualitative studies are, in principle, supportive of one another. The included studies demonstrate a small or larger effect, irrespective of the underlying technological concept employed and the qualitative findings revealed generally positive experiences with telerehabilitation among patients, therapists, and caregivers. The content analysis of the qualitative studies corroborates these findings. It is notable that these positive outcomes may be attributed to a paucity of complexity and openness in the qualitative studies included in this review. When content analysis is limited to reporting mainly positive aspects, it becomes impossible to unravel the complex causal interactions and to identify and address new thematic complexes.

### 4.4. Implications for Practice

The findings support the potential of telerehabilitation as a viable option for stroke rehabilitation, particularly when in-person therapy access is limited. The implementation of effective interventions requires consideration of several key factors. Firstly, interventions must be tailored to the individual needs and capabilities of the patient. Secondly, a variety of engaging activities should be incorporated to maintain motivation. Thirdly, adequate technical support and user training/education must be ensured. Fourthly, social interaction and support should be facilitated, even in remote settings. Finally, performance tracking and feedback mechanisms should be utilized.

### 4.5. Limitations and Future Directions

This review has several limitations. The heterogeneity in intervention types, outcome measures, and study designs makes direct comparisons challenging. Many studies had small sample sizes, limiting generalizability. Additionally, long-term follow-up data was limited in most studies.

Future research should focus on larger, well-designed randomized controlled trials as well as observational and qualitative studies. Especially in qualitative studies, the approach should be open, report negative cases and show new concepts that lead to a better understanding of the complex causal interactions and barriers and facilitators of home-based telerehabilitation in stroke patients.

Future research should also prioritize the standardization of outcome measures to facilitate comparisons, the investigation of long-term effects and adherence, the exploration of hybrid models combining in-person and telerehabilitation approaches, and the development and evaluation of more sophisticated adaptive systems that can easily tailor interventions to individual patient progress.

## 5. Conclusions

In conclusion, the heterogeneity of synchronous telerehabilitation studies to date led to heterogenous effects for subacute and chronic patients, as well as for mild-moderately and moderate-severely affected patients. While telerehabilitation shows promise for improving upper extremity function in stroke patients, further research is needed to optimize intervention designs and understand their long-term impact.

There are aspects mentioned in the qualitative studies that are not explored in the quantitative studies. Qualitative studies extend the conceptual framework beyond the domains of “physical functioning” and “motivation/adherence”. All of the qualitative studies included in this analysis specifically addressed the issue of potential barriers and facilitators to implementation. The most frequently cited barriers are the initial challenges associated with utilizing the technology and the spatial requirements for some of the equipment necessary for telerehabilitation. These factors influence the acceptance of the intervention by both patients and caregivers. Other concepts identified in the qualitative studies include “effort expectancy”, “usability”, and “performance expectancies”, which are not examined in detail in this Pyramid Review.

The aggregated results and findings presented in this Pyramid Review warrant further investigation for several reasons.

Despite a comprehensive search strategy, no observational studies could be found, included, and analyzed in our Pyramid Review. In therapeutic-experimental studies, ideal conditions are created that generally do not correspond to the conditions of subsequent routine healthcare. Unfortunately, therapeutic-observational studies that investigate treatment effects under the conditions of everyday routine healthcare could not be found and included for our research question. The aim of such studies is to change everyday healthcare as little as possible, so that the role of the researcher is ideally limited to collecting and analyzing data.

The transferability of the measured effects from experimental studies to everyday therapeutic practice and for the treatment of stroke patients is constrained by the fact that the results originate from studies conducted in standardized and idealized contexts.

It is evident that there is a paucity of both quantity and diversity in the qualitative studies. The perspectives of subacute patients and their therapists were only collected for one image-based intervention. The effectiveness of sensor-based and VR-based services, as well as the perspectives of caregivers, could not be obtained. The perspectives of chronically affected patients, their therapists, and caregivers were the only ones expressed with regard to sensor-based interventions. Moreover, the qualitative studies included and evaluated in this research do not represent case-oriented and complex findings, resulting in a low quality of results.

The qualitative and quantitative studies fail to address several important issues, including the question of the “ideal” dose, the optimal duration of intervention, and the ideal time since stroke for the implementation of telerehabilitation.

## Figures and Tables

**Figure 1 healthcare-13-00090-f001:**
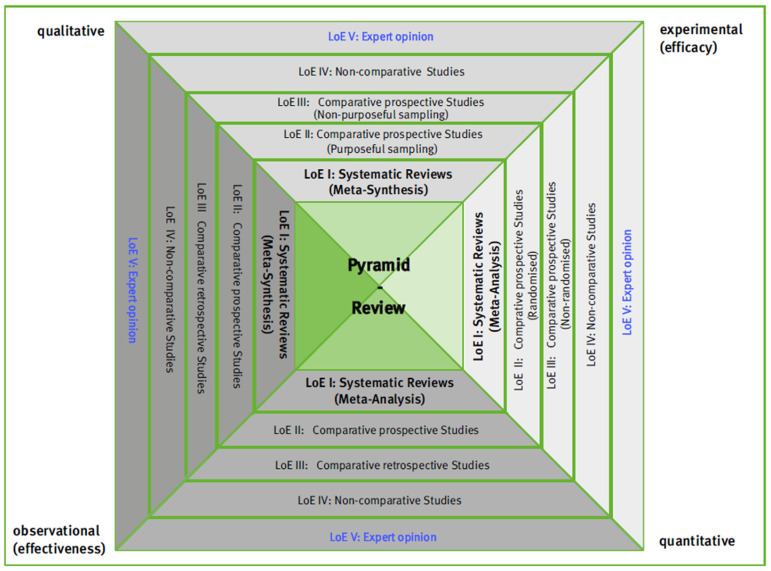
Pyramid Review based on [11]; Note: LoE = Level of Evidence.

**Figure 2 healthcare-13-00090-f002:**
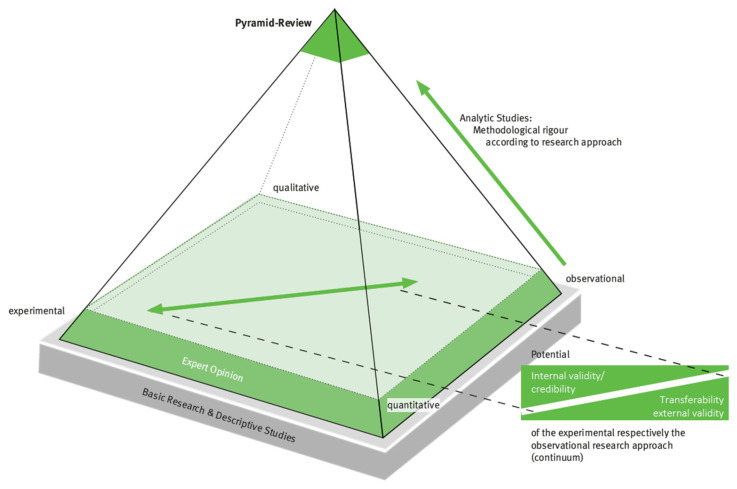
The Research Pyramid based on [16].

**Figure 3 healthcare-13-00090-f003:**
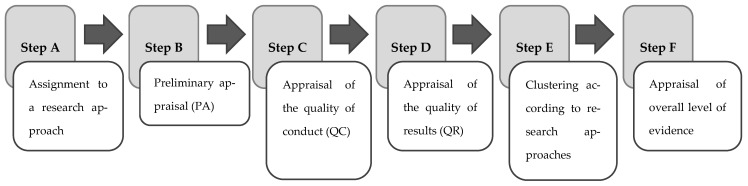
Pyramid Review: Steps of conduct.

**Figure 4 healthcare-13-00090-f004:**
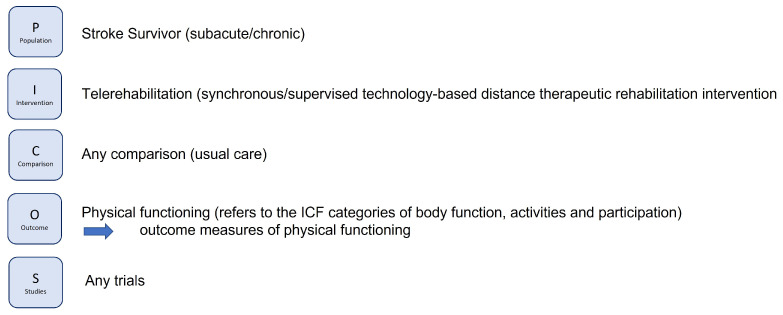
PICOS.

**Figure 5 healthcare-13-00090-f005:**
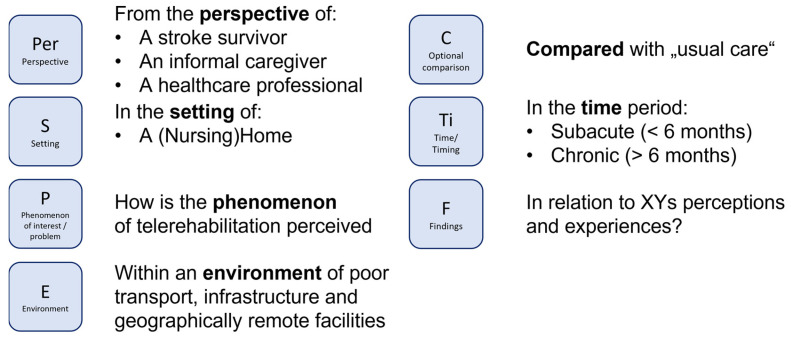
PerSPECTiF.

**Figure 6 healthcare-13-00090-f006:**
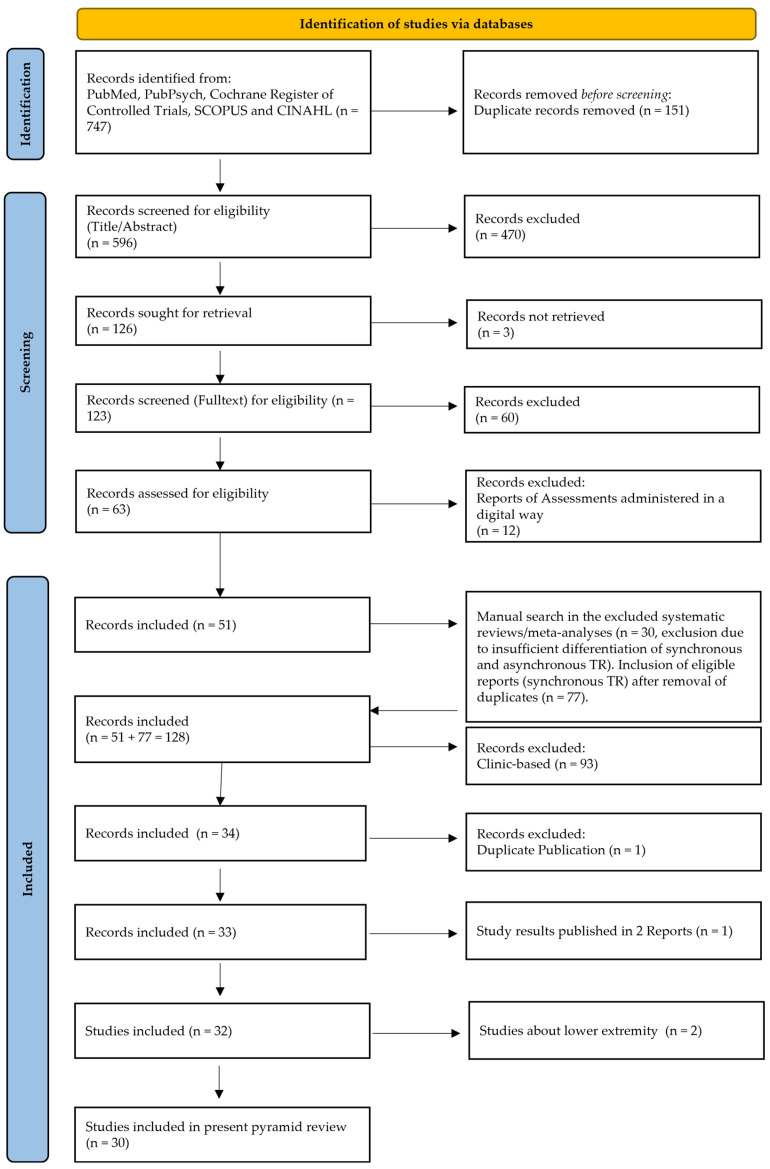
PRISMA Flowchart.

**Figure 7 healthcare-13-00090-f007:**
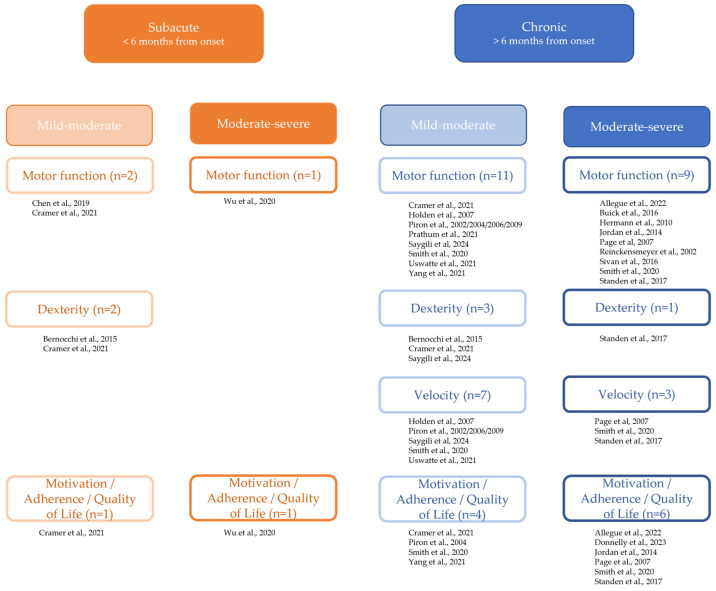
Results: categorization of effects [25,26,27,28,29,30,31,32,33,34,35,36,37,38,39,40,41,42,43,44,45,46,47,48,49,50,51,52].

**Figure 8 healthcare-13-00090-f008:**
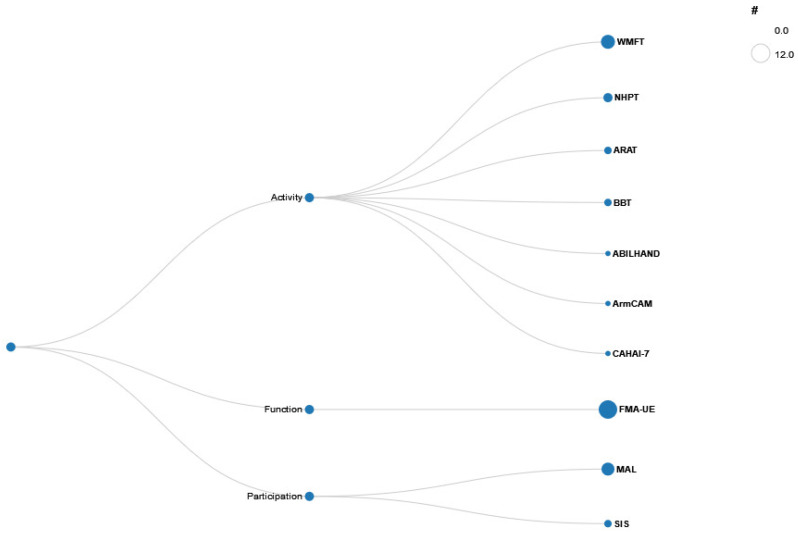
Outcome Measures according to the ICF. Note: WMFT = Wolf Motor Function Test; NHPT = Nine Hole Peg Test; ARAT = Action Research Arm Test; BBT = Box and Block Test; ArmCAM = Arm Capacity Measure; CAHAI = Chedoke Arm and Hand Activity Inventory; FMA-UE = Fugl–Meyer Assessment; MAL = Motor Activity Log; SIS = Stroke Impact Scale; # = quantity of occurrence.

**Figure 9 healthcare-13-00090-f009:**
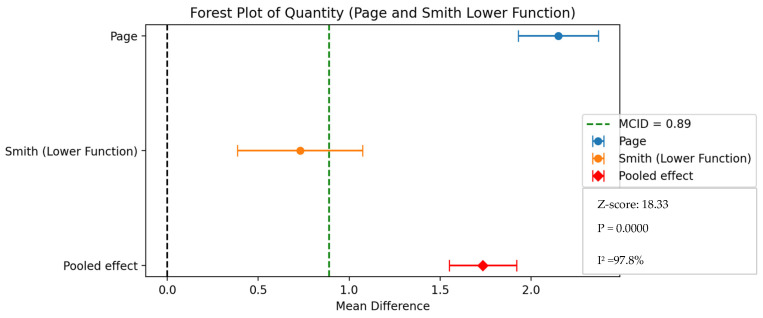
Motor Activity Log (Amount of Use; AOU): effects in sensor-based CIMT studies for moderate-severe affected stroke patients.

**Figure 10 healthcare-13-00090-f010:**
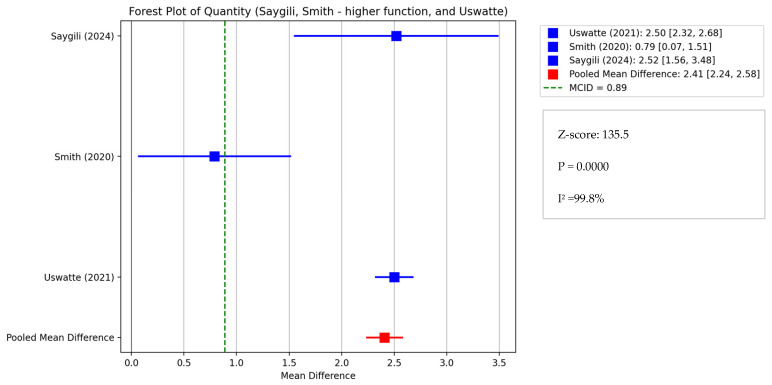
Motor Activity Log (Amount of Use; AOU): effects in sensor-based CIMT studies for mild-moderate affected stroke patients [33,34,44].

**Figure 11 healthcare-13-00090-f011:**
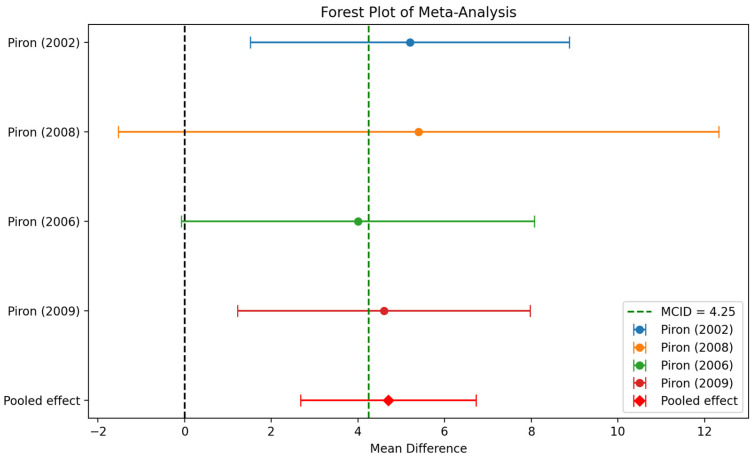
Forest plot of effects in Piron et al. studies [26,27,40,41].

**Table 1 healthcare-13-00090-t001:** Subacute Studies.

	**Wu, 2020 (IG) ^1^ [32]**	**Bernocchi, 2015 [34]**	**Cramer, 2021 [36]**
n	30;	13	8
months since stroke; mean (SD)	<6	3.73 (1.3)	2.7 (1.6)
Degree of Impairment	severe	-	moderate-mild
Intervention	Transition of an early in-hospital rehabilitation program to a home-based setting through the implementation of a collaborative remote rehabilitation nursing model.	Transition of an inpatient rehabilitation protocol to the home environment with an image-based home monitoring and rehabilitation (HBTR) program.	Complex approach to rehabilitation care: 15 min of functional games, at least 15 min of exercises, and 5 min of stroke education using a Jeopardy-style game.
Administered by	Rehabilitation therapists, nurse, caregivers	Physical therapist, nurse, caregivers	Licensed occupational therapist or physical therapist
Underlying Technology	image-based	image-based	VR ^2^-based
Outcome Measure for physical functioning (UE)	FMA-UE	NHPT ^3^	FMA-UE ^4^; BBT ^5^
Dose of Intervention	12 weeks; 2×/week	12 weeks; 3×/week	12 weeks; 6 × 60 min/week + free play
Effects: Cohen’s d, (+), (∅), (−)	motor function (1.29); quality of life (+)	dexterity (1.22); dependency degree (+)	motor function (0.19); dexterity (4.63); motivation/adherence (−)
Level of Evidence	(PA) IV; (QC) IV; (QR) III	(PA) IV; (QC) IV; (QR) III ^6^	(PA) IV; (QC) IV; (QR) IV

^1^ Intervention Group, ^2^ Virtual Reality, ^3^ Hole Peg Test; ^4^ Fugl–Meyer Assessment Upper Extremity, ^5^ Box and Block Test, ^6^ PA = preliminary appraisal, QC = quality of conduct, QR = quality of results; Effects: (+) = big effect, (∅) = medium effect, (−) = low effect.

**Table 2 healthcare-13-00090-t002:** Image-based studies for chronic patients.

	**Bernocchi, 2015 [34]**	**Yang, 2021 [46]**
n	10	9
months since stroke; mean (SD)	15.7 (4.8)	65.86 (111.15)
Degree of Impairment	-	moderate-mild
Intervention	Transition of an inpatient rehabilitation protocol to the home environment with an image-based home monitoring and rehabilitation (HBTR) program.	Virtual Graded Repetitive Arm Supplementary Program (GRASP) delivered via videoconferencing.
Administered by	Physical therapist, nurse, caregivers	Occupational therapist
Underlying Technology	image-based	image-based
Outcome Measure for physical functioning (UE)	NHPT ^1^	ArmCAM ^2^, SIS ^3^
Dose of Intervention	12 weeks; 3×/week	10 weeks; 1 × 60 min/week group intervention; 5 × 60 min individual self-administered training
Effects: Cohen‘s d, (+), (∅), (−)	dexterity (0.64); independence (+);	motor function (0.16); hand function (0.56); adherence (+)
Level of Evidence	(PA) IV; (QC) IV; (QR) IV ^4^	(PA) III; (QC) III; (QR) III

^1^ Hole Peg Test; ^2^ Arm Capacity Measure, ^3^ Stroke Impact Scale, ^4^ PA = preliminary appraisal, QC = quality of conduct, QR = quality of results; Effects: (+) = big effect, (∅) = medium effect, (−) = low effect.

**Table 3 healthcare-13-00090-t003:** Sensor-based FES studies for chronic patients.

	**Prathum, 2021 (IG) ^1^ [29]**	**Buick, 2016 [35]**	**Hermann, 2010 [37]**
n	12	11	1
months since stroke; mean (SD)	16.33 (3.30)	51.64 (47.03)	44
Degree of Impairment	moderate-mild	severe-moderate	severe-moderate
Intervention	Dual-hemispheric transcranial direct current stimulation (dual-tDCS) using an electrode cap with a specified location of the M1 upper limb motor area combined with exercise on motor performance.	Game-based intensive, task-oriented exercise therapy (ET) on a passive workstation (ReJoyce) instrumented with sensors detecting displacement. Patient wore a wristlet for functional electrical stimulation (FES).	FES–Ness H200 Hand Rehabilitation System (surface neuromuscular electrical stimulation (NMES) + wrist extension orthesis) for individualized therapy sessions (increasing affected upper-extremity use during ADLs).
Administered by	Physical therapist	Physical therapist	Occupational therapist
Underlying Technology	sensor-based	sensor-based	sensor-based
Outcome Measure for physical functioning (UE)	FMA-UE, WMFT ^3^	ARAT	FMA-UE ^2^, ARAT ^4^
Dose of Intervention	4 weeks; 3 × 60 min/weeks + 3 × 20 min/week dual tDCS	6 weeks; 5 × 60 min/week	3 weeks; 2 × 30 min/week + 3 × 60 min/week unsupervised
Effects: Cohen‘s d, (+), (∅), (−)	motor function (1.47)	hand function (5.8), adherence (+)	motor function (+), hand function (+)
Level of Evidence	(IA) II; (QI) II; (QR) II	(PA) IV; (QC) IV; (QR) III	(PA) IV; (QC) V; (QR) V ^5^

^1^ Intervention Group, ^2^ Fugl–Meyer Assessment Upper Extremity, ^3^ Wolf Motor Function Test, ^4^ Action Research Arm Test, ^5^ PA = preliminary appraisal, QC = quality of conduct, QR = quality of results; Effects: (+) = big effect, (∅) = medium effect, (−) = low effect.

**Table 4 healthcare-13-00090-t004:** Sensor-based CIMT studies for chronic patients.

	**Saygili, 2024 (IG)** [33]	**Uswatte, 2021 (IG) ^1^** [34]	**Smith, 2020** [44]	**Page, 2007** [47]
n	10	10	28	4
months since stroke; mean (SD)	9.54 (10.9)	31.2 (21.12)	37.2 (4.8)	69.25 (61.4)
Degree of Impairment	moderate-mild	moderate-mild	n = 15: moderate-mild; n = 13: moderate-severe	moderate-severe
Intervention	mCITE: 3–5 ADL activies selected to be used in shaping and task practice sessions + self-administered training of 10 basic exercises (home exercise training)	tele AutoCITE: motor training at the working station + education for behavioral change	iCIMT: group sessions: social interacting + group activities (food preparing, games, simulated iADL); individual sessions: gross and fine motor exercises to encourage transfer of tasks to participants’ ADLs/iADLs	mCITE: 30-min structured functional practice sessions
Administered by	Physical therapist	Occupational/physical therapist	Occupational therapist	Occupational therapist
Underlying Technology	image-based	image-based + sensor-based	image-based	image-based
Outcome Measure for physical functioning (UE)	MAL, WMFT, FMA-UE, NHPT	MAL, WMFT	MAL, WMFT, FMA-UE ^4^	MAL ^2^, WMFT ^3^
Dose of Intervention	3 weeks; 5 × 90 min/week + 3 weeks; 5 h/weekday mitt	2 weeks; 5 × 210 min/week + 2 weeks (35 h) + 2 weeks; 90% of daytime mitt	6 weeks; 1 × 60–90 min/week group session (Lab); 2 × 60 min/week individual therapy (21 h) + 6 weeks; 4 h/day mitt	10 weeks; 3 × 30 min/week (15 h) + 10 weeks; 5 h/day mitt
Effects: Cohen’s d; (+), (∅), (−)	motor function (5.32); amount of use (2.29); quality of use (2.24); velocity (3.25); dexterity (2.09)	motor function: amount of use (12.5); velocity (0.30); satisfaction (+)	motor function (0.54/0.40); amount of use (0.78/1.16); quality of use (0.7/1.06); velocity (0.25/0.16); education for patient + family (+); adherence (+)	motor function: amount of use (9.56); quality of use (11.24); velocity (1.59); adherence (+)
Level of Evidence	(PA) IV; (QC) IV; (QR) IV	(PA) IV; (QC) IV; (QR) III	(PA) IV; (QC) IV; (QR) IV	(PA) IV; (QC) IV; (QR) IV ^5^

^1^ Intervention Group, ^2^ Motor Activity Log, ^3^ Wolf Motor Function Test, ^4^ Fugl–Meyer Assessment Upper Extremity, ^5^ PA = preliminary appraisal, QC = quality of conduct, QR = quality of results; Effects: (+) = big effect, (∅) = medium effect, (−) = low effect.

**Table 5 healthcare-13-00090-t005:** Sensor-based studies for chronic patients by Piron et al.

	**Piron, 2006 (IG) [26]**	**Piron, 2009 (IG) [27]**	**Piron, 2004/2008 (IG) ^1^ [28,41]**	**Piron, 2002 [40]**
n	12	18	5	5
months since stroke; mean (SD)	11.8 (4.1) (IG + CG) ^2^	14.7 (6.6)	12.8 (1.9)	17 (15.8)
Degree of Impairment	moderate-mild	moderate-mild	moderate-mild	mild
Intervention	VRRS.net: 3D motion tracking device connected to an object or a glove worn by the patient + videoconferencing: motor skills which resemble ADLs	VRRS.net: 3D motion tracking device connected to an object or a glove worn by the patient + videoconferencing: motor skills which resemble ADLs	workstation with a 3D tracking device + videoconferencing. System provided visual feedback involved in motor learning, i.e., knowledge of performance of UL motor tasks and of results.	VRRS.net: 3D motion tracking device connected to an object or a glove worn by the patient + videoconferencing: motor skills which resemble ADLs
Administered by	Physical therapist	Physical therapist	Physical therapist	Physical therapist
Underlying Technology	image-based + sensor-based	image-based + sensor-based	image-based + sensor-based	image-based + sensor-based
Outcome Measure for physical functioning (UE)	FMA-UE, Abilhand	FMA-UE, Abilhand	FMA-UE	FMA-UE ^3^
Dose of Intervention	4 weeks; 5 × 60 min/week	4 weeks; 5 × 60 min/week	4 weeks; 5 × 60 min/week	6 weeks; 5 × 60 min/week
Effects: Cohen‘s d; (+), (∅), (−)	motor function (0.63); arm velocity (+)	motor function (0.61); arm velocity (+)	motor function (0.63); motivation -> satisfaction (+)	motor function (0.81); arm velocity (+)
Level of Evidence	(PA) IV; (QC) V; (QR) V	(PA) IV; (QC) V; (QR) V	(PA) II; (QC) III; (QR) III	(PA) IV; (QC) V; (QR) V ^4^

^1^ Intervention Group, ^2^ Control Group, ^3^ Fugl–Meyer Assessment Upper Extremity, ^4^ PA = preliminary appraisal, QC = quality of conduct, QR = quality of results; Effects: (+) = big effect, (∅) = medium effect, (−) = low effect.

**Table 6 healthcare-13-00090-t006:** Sensor-based exergames studies for chronic patients.

	**Allegue, 2022 (IG) ^1^ [25]**	**Standen, 2017 (IG) [30]**	**Jordan, 2014 [39]**	**Reinkensmeyer, 2002 [43]**
n	4	13	12	1
months since stroke; mean (SD)	96 (24)	17.2 (34.8)	12.8 (11.7)	15
Degree of Impairment	severe-moderate	severe-moderate	severe-moderate	severe
Intervention	VirTele: Kinect camera + Jintronix exergames for UE rehabilitation + Reacts app to conduct videoconference sessions with clinicians; Motivational interviewing	Exergaming wearing a virtual glove: handmounted power unit, with four infrared light-emitting diodes mounted on the user’s fingertips. The diodes were tracked using one or two Nintendo Wiimotes mounted by the monitor on which the games were displayed to translate the location of the user’s hand, fingers, and thumb in 3D.	Exergaming using an arm skate (Smart Skate) composed of a tray that supports the lower arm, with straps to hold the arm in place for gravity support of the affected arm; motivation questionnaire	Java Therapy system: tracking arm movements through the use of a force feedback joystick, clip-on splint, and armrest allowing movement of the hand in a 10 × 10 cm workspace in the horizontal plane; playing exergames; user satisfaction survey
Administered by	Physical therapist	Physical/Occupational therapist	Physical therapist	Physical/Occupational therapist
Underlying Technology	image-based + sensor-based	image-based + sensor-based	image-based + sensor-based	image-based + sensor-based
Outcome Measure for physical functioning (UE)	FMA-UE ^2^, MAL ^3^, SIS ^4^	WMFT ^6^, NHPT ^7^, MAL	FMA-UE	CAHAI ^5^
Dose of Intervention	8 weeks; 5 × 30 min/week	8 weeks; 7 × 60 min/week	4–6 weeks; 9 or 16 h	12 weeks; 3 × 20 trials/week
Effects: Cohen‘s d; (+), (∅), (−)	motor function (0.00), quantity of use (0.81), quality of use (1.09); quality of life (0.15), motivation (∅)	motor function (0.04), velocity (0.28); quantity of use (0.13): quality of use (0.32) ADL (−), motivation -> adherence (∅)	motor function (1.23); motivation (+)	motor function (∅); user satisfaction (+)
Level of Evidence	(PA) IV; (QC) IV; (QR) IV ^8^	(PA) II; (QC) II; (QR) II	(PA) IV; (QC) IV; (QR) IV	(PA) IV; (QC) V; (QR) V

^1^ Intervention Group, ^2^ Fugl–Meyer Assessment Upper Extremity, ^3^ Motor Activity Log, ^4^ Stroke Impact Scale, ^5^ Chedoke Arm and Hand Activity Inventory, ^6^ Wolf Motor Function Test, ^7^ Nine Hole Peg Test, ^8^ PA = preliminary appraisal, QC = quality of conduct, QR = quality of results; Effects: (+) = big effect, (∅) = medium effect, (−) = low effect.

**Table 7 healthcare-13-00090-t007:** VR-based studies for moderate-mild affected chronic patients.

	**Cramer, 2021 [36]**	**Holden, 2007 [38]**
n	5	11
months since stroke; mean (SD)	24.5 (19.13)	45.6 (37.2)
Degree of Impairment	moderate-mild	moderate-mild
Intervention	Holistic approach to rehabilitation care: 15 min of functional games, at least 15 min of exercises, and 5 minof stroke education using a Jeopardy-style game.	virtual environment-based (VE) telerehabilitation system to conduct interactive VE treatment sessions for improving upper extremity function
Administered by	licensed occupational therapist or physical therapist	physical therapist
Underlying Technology	VR-based ^1^	VR-based
Outcome Measure for physical functioning (UE)	FMA-UE ^2^, BBT ^3^	FMA-UE, WMFT ^4^
Dose of Intervention	12 weeks; 6 × 60 min/week	6 weeks; 5 × 60 min/week
Effects: Cohen‘s d; (+), (∅), (−)	motor function (0.55); dexterity (0.97); motivation/adherence (−);	motor function (0.59); velocity (0.50)
Level of Evidence	(PA) IV; (QC) IV; (QR) IV ^5^	(PA) IV; (QC) V; (QR) V

^1^ Virtual Reality, ^2^ Fugl-Meyer Assessment Upper Extremity, ^3^ Box and Block Test, ^4^ Wolf Motor Function Test, ^5^ PA = preliminary appraisal, QC = quality of conduct, QR = quality of results; Effects: (+) = big effect, (∅) = medium effect, (−) = low effect.

**Table 8 healthcare-13-00090-t008:** Characteristics of qualitative studies.

	**Page, 2007 (MM ^1^)** [47]	**Standen, 2015 (MM)** [48]	**Chen, 2019** [49]	**Donnelly, 2023** [50]	**Maddahi, 2021** [51]	**Sivan, 2016** [52]
Methodology	Case Study	Cohort Study	Case Study	Case Study	Expert Round	Cohort Study
Method	informal interview	semi-structured interviews	semi-structured interviews	phenomenological interviews	focus group	semi-structured interviews
Phenomenon of interest	improving motor changes in valued activities; satisfaction -> adherence	improving upper limb function; perceived barriers + facilitators of exergames	improvements in limb functions, emotional well-being; ease of use -> adherence	improving quality of life, motor control of the hemiparetic arm; acceptance	improving hand function; therapists’ perspectives on designing a new portable hand telerehabilitation platform (PHTP)	improving upper limb function; users’ perspectives and suggestions on the future development
Setting	laboratory	home	home or laboratory	laboratory	online/teleconference	laboratory
Geographical region	USA	Great Britain	USA	USA	Canada/USA	Great Britain
Cultural Context	North American	European	North American	North American	North American	European
Participants	four chronic patients who had completed a 10-week sensor-based telerehabilitation (mCITE) + 1 OT	13 chronic patients who had completed an 8-week sensor-based telerehabilitation	13 subacute patients who had completed a 6-week image-based telerehabilitation + nine caregivers	four chronic patients who had completed a 6-week sensor-based telerehabilitation	four OTs ^2^, one PT ^3^, two engineers, one computer scientist, and the principal investigator	17 patients who provided feedback on the usability and impact of a robotic-based device post-use (end-users) + seven OTs/PTs (professional users)
Data analysis	n.a. ^4^	Thematic analysis	Thematic analysis	Thematic analysis	Thematic analysis	Content analysis

^1^ Mixed-methods study, ^2^ Occupational Therapist, ^3^ Physical Therapist, ^4^ not available.

**Table 9 healthcare-13-00090-t009:** Level of Evidence (qualitative studies).

	**Page, 2007** [47]	**Standen, 2015** [48]	**Chen, 2019** [49]	**Donnelly, 2023** [50]	**Maddahi, 2021** [51]	**Sivan, 2016** [52]
preliminary appraisal (PA)	IV	IV	IV	IV	V	IV
Quality of conduct (QD)	poor	good	good	good	medium	medium
Level of Plausibility	unsupported	unequivocal	unequivocal	unequivocal	unequivocal	unequivocal
Quality of result (QR)	Excluded	IV	IV	IV	V	IV

## Data Availability

All research data for this Pyramid Review is included in the Appendix A.

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
