# Peer review of "Synchronous Home-Based Telerehabilitation of the Upper Extremity Following Stroke—A Pyramid Review"

_healthcare, 2025, doi:10.3390/healthcare13010090_

Round 1
Reviewer 1 Report (Previous Reviewer 1)
Comments and Suggestions for Authors
I have no comments on this manuscript. All recommendations have been implemented
Author Response
Thank you very much. I hope our review will be published soon.
Reviewer 2 Report (New Reviewer)
Comments and Suggestions for Authors
Comments:
This a systematic review evaluating synchronous homebased telerehabilitation of the upper extremity following stroke. Authors have used pyramid review methodology and have included 30 studies in this review. They report home based rehabilitation to be associated with improving upper extremity function in stroke patients. The method of pyramid review is a relatively less used method. Authors have discussed the benefits of such review and provide insight into the methods of it in the introduction part. Regarding the research question, it is not novel but the evaluation of evidence in a pyramid review method is interesting. The manuscript is logically structured and easy to read. The authors need to address the following points:
1- The manuscript attached is not clean. It has highlighted portions; I am not sure what does it indicate. It seems like a pre-final version.
2- While discussing the Pyramid review, authors may give a brief description how it started. The steps in the methodology of such review should be supplemented by references. (The main references are in German)
3- The reporting of such reviews usually follows the PRISMA guidelines, which mandates prior registration at a database. Authors may clarify if this study was registered or not.
4- The PRISMA flow diagram should indicate the number of searches obtained from each database.
5- Results: 32 studies were included in the review. But the studies add up to 32 (24+4+2) (Line 282-284).
6- Results: 3.2.1- ‘All participants were adults, comprising 351 therapist and 354 patients’. Were therapists included as study participants? If so which studies considered the same and how the outcome measures were assessed in these studies may be mentioned.
7- The assessment of bias for the included studies have not been mentioned. Similarly, issue of publication bias should d also be addressed.
8- The forest plot should mention the outcome measure of interest.
Author Response
Please see the attachement.

Reviewer 3 Report (New Reviewer)
Comments and Suggestions for Authors
Dear Authors,
The study addresses a highly relevant and current topic, and the methodology employed is robust and well-described. However, I would like to offer some suggestions for improvement and pose some questions that could help clarify certain aspects of the manuscript.
Introduction. The introduction is consistent and presents the topic clearly. It describes key concepts and provides relevant data. The bibliography used adequately supports the article's approach. The objective is coherent and appropriate for the type of review proposed. However, the objective should be included at the end of this section.
Methodology. I have some questions regarding this section.
Why did you choose the Pyramid review methodology over more traditional ones? Have you considered conducting a subgroup analysis based on stroke severity to see if there are differences in the effectiveness of the interventions? Could you provide more details on how adherence to telerehabilitation interventions was measured and what factors influenced it?
The section dedicated to explaining the pyramid review methodology is very detailed. While it is important to provide a clear understanding of the methodological approach, I believe it could be beneficial to simplify this section. You could summarize the key points of the methodology and redirect the focus towards the results obtained and their practical implications. This will allow readers to focus on the findings and their clinical relevance.
To increase the transparency and credibility of the review process, it would be helpful to explicitly mention if any recognized checklist, such as PRISMA or STROBE, was used to assess the quality of the included studies.
Discussion. In addition to mentioning the limitations, it would be useful to add the strengths of the study. This would help better contextualize the findings and their relevance.
Additional Comments.
The article is difficult to read in some sections. It would be beneficial to review and simplify these sections to improve clarity and accessibility. It is also unclear why there are sections highlighted in yellow in the manuscript. It would be helpful to provide an explanation or remove the highlighting if it is not necessary.
The bibliography does not follow the journal's guidelines. I recommend the authors review this section and adjust the references to meet the specific requirements of the journal.
The supplementary material included in the manuscript is not clear or easy to understand. I suggest reviewing it, improving the presentation format, and simplifying the content, ensuring that each section is clearly labeled and explained.
I hope this helps.
Author Response
Please see the attachment

Reviewer 4 Report (New Reviewer)
Comments and Suggestions for Authors
-
Clarity of Objectives:
- The research question and objectives are clear and address an important clinical topic. However, consider emphasizing the significance of telerehabilitation for global healthcare settings, especially in underserved areas.
- Align the clinical question more prominently with the conclusions drawn to ensure coherence throughout the manuscript.
-
Methodology:
- The methodology is thorough, particularly the Pyramid Review approach, which is innovative. However, more detail on the challenges faced during study categorization (e.g., discrepancies in quality appraisals) could enhance transparency.
- Consider including a flowchart or diagram to better explain the Pyramid Review methodology for readers unfamiliar with this approach.
-
Scope and Literature Inclusion:
- The review successfully integrates quantitative and qualitative studies, but the exclusion of observational studies is a limitation. Could the authors discuss potential biases introduced by this gap?
- Expand the explanation of the PerSPECTiF principle in the methods to enhance comprehension for readers outside the domain.
Round 2
Reviewer 2 Report (New Reviewer)
Comments and Suggestions for Authors
The authors have addressed my previous queries adequately and necessary changes have been made in the manuscript.
Author Response
Thank you very much!
Reviewer 3 Report (New Reviewer)
Comments and Suggestions for Authors
The authors have addressed all the suggestions made. However, the references still do not comply with the journal's guidelines. I recommend their review.
Best regards
Author Response
Thank you for your comment.
Changes have been made to comply with the journals guideline
This manuscript is a resubmission of an earlier submission. The following is a list of the peer review reports and author responses from that submission.
Round 1
Reviewer 1 Report
Comments and Suggestions for Authors
The manuscript reports on a systematic review of 29 randomized controlled trials that presented the use of telerehabilitation techniques for stroke patients. This topic is both relevant and promising, as the authors have thoroughly detailed.
The results in this paper clearly indicate that tele-rehabilitation technologies are effective in restoring upper limb function post-stroke.
The manuscript is well-written and adheres to solid methodological standards. The data and conclusions are likely to be of interest to researchers in neurology, rehabilitation, etc., making this research suitable for publication in the journal Healthcare.
However, I recommend significantly shortening the "Introduction" section, as it contains an overly detailed description of the methodology for constructing a research pyramid.
Author Response
Comment: I recommend significantly shortening the "Introduction" section, as it contains an overly detailed description of the methodology for constructing a research pyramid.
Answer: Thank you very much for your kind recommendation. Please find the significantly shortened introduction in the newly uploaded manuscript
Reviewer 2 Report
Comments and Suggestions for Authors
Stangenberg-Gliss et al., 2024 present a paper titled Synchronous Homebased Telerehabilitation of the Upper Extremity following Stroke – a Pyramid review. My comments of this paper appear below:
1. What is the logic behind pages 3-10 being simply a drawn-out, wordy description of what appears to be a meta-analysis? If there is something completely novel about this then you should have a comparison between Your design and a systematic review Vs a meta-analysis or a covidence study. You cite 1 paper which has used a similar method ie pyramid however, this was published in 2011 (13 years ago), then you cite a book that discusses the method [78] only but does not use it, and [79] who transparently states that they are publishing a systematic review.
2. Is your publication about the methods or telerehabilitation of stroke suffers? The methods are so difficult to read (and are repetitive) that by the time I got to the data/real results the rationale behind your work was forgotten.
3. Figure 6 Page 11 should be in the methods, along with a couple of paragraphs (not 7 pages) – then your results should be a table explaining what you found in association with the research question.
4. Page 13 lines 467 onwards when you are explaining each of the tests/diagnostic tools used to generate an outcome. There are no references to these tests, should the studies or people whose intellectual property they belong to be referenced here? You have only one, line 570 page 15 [64].
5. Your results could be clearer, and more succinct in the presentation. There is a lot of valuable information, however, it required jumping forward and back to see where the current data fitted in. I recommend a result flow chart of the main findings in the results section. Which makes it clear from the onset that there are 5 categories of telerehab covered and what the findings were - Improved symptoms or not improved ect. In the position that figure 6 is in. This would allow the reader to just flick to this flowchart when reading the discussion.
Author Response
Comments 1: What is the logic behind pages 3-10 being simply a drawn-out, wordy description of what appears to be a meta-analysis? If there is something completely novel about this then you should have a comparison between Your design and a systematic review Vs a meta-analysis or a covidence study. You cite 1 paper which has used a similar method ie pyramid however, this was published in 2011 (13 years ago), then you cite a book that discusses the method [78] only but does not use it, and [79] who transparently states that they are publishing a systematic review.
Answer: The logic behind pages 3-10 was (we have shortened them according to the respective recommendations) to explain the method of a pyramid review, which has not yet been described in english. Meta-analyses are only one part of this approach. What is completely new is that experimental studies, which usually investigate efficacy under standardized/idealized study conditions and have their strength in internal validity, are reviewed separately from observational studies, which usually investigate effectiveness under routine care conditions and have their strength in external validity. Thus, a pyramid review can provide insights about the potential of interventions and the possibility of realizing this potential in everyday care. In addition, the same review approach integrates qualitative research, but not generelly, as it is common in mixed method-reviews, but rather with a clear focus on perceived effectiveness and on possible decoding and explanation of complex causal interactions. The results of the individual partial reviews are interrelated to one each other in order to obtain a comprehensive understanding of the overall state of the evidence.
In fact we cite a paper from 2011 that presented the methodological foundations of the research pyramid for the first time in an English-language journal [72]. The elaboration and testing in pilot reviews of details and operationalizations of this comprehensive approach took longer than we hoped. The book chapter we cited was then the first more detailed description on how to conduct a pyramid review in 2022 [9], and the journal-article also published in 2022 describes an appraising-tool developed especially for pyramid reviews. Also in 2022 the first comprehensive pyramid review was funded by German Federal Ministry of Education and Research, grant 1146 number 03FHP179. The submitted manuscript now summarizes the results of this pyramid review.
Comments 2: Is your publication about the methods or telerehabilitation of stroke suffers? The methods are so difficult to read (and are repetitive) that by the time I got to the data/real results the rationale behind your work was forgotten.
Answer 2: Our publication is clearly a review about synchronous homebased telerehabilitation of the upper extremity following stroke. Our pyramid review approach is new and therefor we included the methodology. We submitted this part now as a methods paper elsewhere and shortened the methods section in this publication significantly (see page 4-6).
Comments 3: Figure 6 Page 11 should be in the methods, along with a couple of paragraphs (not 7 pages) – then your results should be a table explaining what you found in association with the research question.
Answer 3: In accordance with standard research practice (see also Cochrane Handbook III 3.5.1), we have decided to leave the flowchart for the literature search (figure 6) in the results section (now on page 9). The literature search was updated and the results section now includes another sensor-based publication from 2024. The participants section (line 317 onwards) was updated as well.
Comments 4: Page 13 lines 467 onwards when you are explaining each of the tests/diagnostic tools used to generate an outcome. There are no references to these tests, should the studies or people whose intellectual property they belong to be referenced here? You have only one, line 570 page 15 [64].
Answer 4: Please find the references to each of the tests/assessments from line 345 onwards as well as in the reference list.
Comments 5: Your results could be clearer, and more succinct in the presentation. There is a lot of valuable information, however, it required jumping forward and back to see where the current data fitted in. I recommend a result flow chart of the main findings in the results section. Which makes it clear from the onset that there are 5 categories of telerehab covered and what the findings were - Improved symptoms or not improved ect. In the position that figure 6 is in. This would allow the reader to just flick to this flowchart when reading the discussion.
Answer 5: We have added Figure 7. This figure lists the effects of the individual studies and categorizes them according to subacute/chronic and mild-moderate/moderate-severe affected patients. The classification and comparison of the effect size according to the underlying technology of the intervention (image-, sensor- or VR-based) can still be found in the corresponding chapters.
Reviewer 3 Report
Comments and Suggestions for Authors
The manuscript presents a comprehensive review of the current literature on synchronous home-based telerehabilitation for upper extremity functional impairments in stroke patients. But it seems the paper was classified as a reviewer paper, but I found it more resembles a literature-based research article. The title is confusing, and need to be more clearly stated or described. The manuscript requires re-organized to make it more clear that if this is part of research or just a summarizing review paper. Here are a few concerns I would like to propose if this is a research article.
1. The manuscript mentions a narrative synthesis due to the heterogeneity of the included studies. Can the authors elaborate on how they categorized the studies for this synthesis and the criteria used to evaluate the subgroups?
2. The authors highlight differences between quantitative and qualitative studies. How do the authors plan to integrate findings from qualitative studies that address barriers and facilitators of telerehabilitation with the quantitative outcomes?
3. Given the limitations in the types of studies included (e.g., lack of observational studies), how do the authors address the generalizability of their findings to broader clinical practice?
4. The manuscript emphasizes the importance of technical support and user training. What recommendations do the authors have for implementing effective training programs for both patients and healthcare providers?
5. The conclusion states that further research is needed to understand the long-term impact of telerehabilitation. What specific aspects of long-term outcomes do the authors believe are most critical to investigate? What is the most significant conclusions the authors want to convey from the search?
6. The qualitative studies identified barriers such as technology challenges and spatial requirements. How do the authors suggest these barriers can be mitigated in future telerehabilitation programs?
Author Response
General comments: The manuscript presents a comprehensive review of the current literature on synchronous home-based telerehabilitation for upper extremity functional impairments in stroke patients. But it seems the paper was classified as a reviewer paper, but I found it more resembles a literature-based research article. The title is confusing, and need to be more clearly stated or described. The manuscript requires re-organized to make it more clear that if this is part of research or just a summarizing review paper. Here are a few concerns I would like to propose if this is a research article.
General answer: This is a review paper concerning synchronous homebased telerehabilitation of the upper extremity following stroke. As method we used the pyramid review approach.
Comments 1. The manuscript mentions a narrative synthesis due to the heterogeneity of the included studies. Can the authors elaborate on how they categorized the studies for this synthesis and the criteria used to evaluate the subgroups?
Answer 1: We categorized the participants into subacute and chronic patients and divided the evaluation and appraisal into mild-moderate and moderate-severe affected patients. To make the subgroups more comparable, we grouped the studies according to the underlying technological concepts (image-based, sensor-based, and VR-based). We have added a new figure (Figure 7) to make it easier to see the categorization and the effects being evaluated. The heterogeneity of the included studies can be seen in the severity of the patients affected, the duration of the interventions and the different outcome measures. Meta-analyses were calculated where possible.
- The authors highlight differences between quantitative and qualitative studies. How do the authors plan to integrate findings from qualitative studies that address barriers and facilitators of telerehabilitation with the quantitative outcomes?
Answer 2: In this review, we evaluated and compared primarily the effects of synchronous telerehabilitation in stroke patients in the included studies. The effects published in qualitative studies (often named perceived effectiveness) provide valuable insights that should be considered in the implementation of telerehabilitation services and/or in observational studies.
Comments 3: Given the limitations in the types of studies included (e.g., lack of observational studies), how do the authors address the generalizability of their findings to broader clinical practice?
Answer 3: Since observational studies are lacking for our research question, the transferability and generalizability of the results is limited. Further research would have to carry out observational studies in order to be able to draw dependable conclusions. Until such results are available, therapists must continue to carefully examine whether results from experimental studies are effective for their own patients in their specific setting and context.
Comments 4. The manuscript emphasizes the importance of technical support and user training. What recommendations do the authors have for implementing effective training programs for both patients and healthcare providers?
Answer 4: This review summarizes the effects of the included studies, but the studies do not provide any specific indications of successful implementation. This would require a separate study with a different research question. We can only provide general implications for practice published in the included studies (from line 950).
Comments 5: The conclusion states that further research is needed to understand the long-term impact of telerehabilitation. What specific aspects of long-term outcomes do the authors believe are most critical to investigate? What is the most significant conclusions the authors want to convey from the search?
Answer 5: In our opinion, it would be particularly interesting to look at the area of participation in the long term. Can the sometimes huge effects achieved in the studies be translated into everyday activities? Do they lead to more independence, participation and quality of life for stroke survivors and their families?
Comments 6: The qualitative studies identified barriers such as technology challenges and spatial requirements. How do the authors suggest these barriers can be mitigated in future telerehabilitation programs?
Answer 6: Despite our focus on effects we found other aspects like barriers and facilitators for the conduct of telerehabilitation. Technological progress is clearly visible in this area and will certainly continue in the future. It is therefore conceivable and desirable that digital health interventions will be feasible in the home and that the technical equipment required to carry out these therapies will take up less space.